# Molecular Docking and Dynamics Simulation Studies Predict Potential Anti-ADAR2 Inhibitors: Implications for the Treatment of Cancer, Neurological, Immunological and Infectious Diseases

**DOI:** 10.3390/ijms24076795

**Published:** 2023-04-05

**Authors:** Emmanuel Broni, Andrew Striegel, Carolyn Ashley, Patrick O. Sakyi, Saqib Peracha, Miriam Velazquez, Kristeen Bebla, Monsheel Sodhi, Samuel K. Kwofie, Adesanya Ademokunwa, Sufia Khan, Whelton A. Miller

**Affiliations:** 1Department of Medicine, Loyola University Medical Center, Loyola University Chicago, Maywood, IL 60153, USA; 2Department of Chemical and Biochemistry, College of Science, University of Notre Dame, Notre Dame, IN 46556, USA; 3Department of Chemistry, School of Physical and Mathematical Sciences, College of Basic and Applied Sciences, University of Ghana, Legon, Accra P.O. Box LG 56, Ghana; 4Department of Chemical Sciences, School of Sciences, University of Energy and Natural Resources, Sunyani P.O. Box 214, Ghana; 5Department of Molecular Pharmacology & Neuroscience, Loyola University Medical Center, Loyola University Chicago, Maywood, IL 60153, USA; 6Department of Biomedical Engineering, School of Engineering Sciences, College of Basic & Applied Sciences, University of Ghana, Legon, Accra P.O. Box LG 77, Ghana; 7Department of Biochemistry, Cell and Molecular Biology, West African Centre for Cell Biology of Infectious Pathogens, College of Basic and Applied Sciences, University of Ghana, Accra P.O. Box LG 54, Ghana; 8Department of Cognitive and Behavioral Neuroscience, Loyola University Chicago, Chicago, IL 60660, USA; 9Department of Biology, Loyola University Chicago, Chicago, IL 60660, USA

**Keywords:** adenosine deaminases acting on RNA (ADAR), anti-ADAR2, natural products, cancer, depression, anxiety disorders, autism spectrum disorder (ASD), molecular docking, molecular dynamics simulation

## Abstract

Altered RNA editing has been linked to several neurodevelopmental disorders, including autism spectrum disorder (ASD) and intellectual disability, in addition to depression, schizophrenia, some cancers, viral infections and autoimmune disorders. The human ADAR2 is a potential therapeutic target for managing these various disorders due to its crucial role in adenosine to inosine editing. This study applied consensus scoring to rank potential ADAR2 inhibitors after performing molecular docking with AutoDock Vina and Glide (Maestro), using a library of 35,161 compounds obtained from traditional Chinese medicine. A total of 47 compounds were predicted to be good binders of the human ADAR2 and had insignificant toxicity concerns. Molecular dynamics (MD) simulations, including the molecular mechanics Poisson–Boltzmann surface area (MM/PBSA) procedure, also emphasized the binding of the shortlisted compounds. The potential compounds had plausible binding free energies ranging from −81.304 to −1068.26 kJ/mol from the MM/PBSA calculations. ZINC000085511995, a naphthoquinone had more negative binding free energy (−1068.26 kJ/mol) than inositol hexakisphosphate (IHP) [−873.873 kJ/mol], an agonist and a strong binder of ADAR2. The potential displacement of IHP by ZINC000085511995 in the IHP binding site of ADAR2 could be explored for possible deactivation of ADAR2. Bayesian-based biological activity prediction corroborates the neuropharmacological, antineoplastic and antiviral activity of the potential lead compounds. All the potential lead compounds, except ZINC000014612330 and ZINC000013462928, were predicted to be inhibitors of various deaminases. The potential lead compounds also had probability of activity (Pa) > 0.442 and probability of inactivity (Pi) < 0.116 values for treating acute neurologic disorders, except for ZINC000085996580 and ZINC000013462928. Pursuing these compounds for their anti-ADAR2 activities holds a promising future, especially against neurological disorders, some cancers and viral infections caused by RNA viruses. Molecular interaction, hydrogen bond and per-residue decomposition analyses predicted Arg400, Arg401, Lys519, Trp687, Glu689, and Lys690 as hot-spot residues in the ADAR2 IHP binding site. Most of the top compounds were observed to have naphthoquinone, indole, furanocoumarin or benzofuran moieties. Serotonin and tryptophan, which are beneficial in digestive regulation, improving sleep cycle and mood, are indole derivatives. These chemical series may have the potential to treat neurological disorders, prion diseases, some cancers, specific viral infections, metabolic disorders and eating disorders through the disruption of ADAR2 pathways. A total of nine potential lead compounds were shortlisted as plausible modulators of ADAR2.

## 1. Introduction

RNA editing is a post-transcriptional process that alters RNA sequences with profound physiological consequences. The most common form of RNA editing in mammalian brain is catalyzed by the adenosine deaminases acting on RNA (ADAR) enzymes, ADAR1 and ADAR2, resulting in adenosine (A) to inosine (I) substitutions in mRNA and microRNAs, which is implicated in various diseases [1,2]. A-to-I editing is involved in regulating the immune system, RNA splicing, protein recoding, microRNA biogenesis, and in the formation of heterochromatin [3]. By catalyzing the conversion of A-to-I, these enzymes frequently alter the sequence of mRNA before translation, and potentially the primary structure of the resultant proteins [4]. Moreover, RNA editing of non-coding RNAs, e.g., microRNAs, can alter microRNA-mediated regulation of gene expression. RNA editing plays a critical role in the development of several organ systems including the brain, and has been associated with the molecular pathophysiology of a broad range of human diseases.

ADAR2-mediated RNA editing activity is crucial for the nervous system functions, thus altered RNA editing has been linked to neurological disorders [2,4,5]. A few studies linked increased RNA editing of the 5-Hydroxytryptamine (or serotonin) 2C receptor (5-HT2CR) to the neurodevelopmental disorder, Prader-Willi syndrome (PWS) [6]. Patients with PWS have a multisystem genetic disorder that includes cognitive disability and a behavioral phenotype with similarities to autism spectrum disorders [6,7]. Indeed, several studies of postmortem brain reveal higher levels of RNA editing in autism patients, specifically for the AMPA-type receptor subunits, kainate-type receptor subunits, NEAT1, CTSB and 5-HT2CR [8]. Other studies have shown that 5-HT2CR RNA editing is increased in the prefrontal cortex of individuals who die by suicide [9,10,11,12,13,14,15,16]. In rodents, increased 5-HT2CR RNA editing results in aggressive and anxiety-like behaviors, and other behaviors that may align with post-traumatic stress disorder [17,18,19,20]. Increased 5-HT2CR RNA editing decreases G-protein coupling of the receptor, thereby reducing its signal transduction and reducing the constitutive activity of the receptor [14,21].

It has been shown that a synthetic helix-threading peptide that binds 5-HT2CR editing sites was able to selectively inhibit ADAR2 editing of 5-HT2C RNA in vitro [22]. However, the bioavailability of such molecules is limited. Small molecule compounds that inhibit 5-HT2CR RNA editing have not been identified and these compounds are potential treatments for neurological illnesses that are triggered or exacerbated by psychological stress. Inhibition of ADAR-mediated RNA editing of the 5-HT2C receptor increases the activity of the 5-HT2C receptor, leading to reduced food intake [23,24,25]. This effect may be therapeutic in metabolic disorders, such as diabetes and in hyperphagia-induced obesity. This may also be therapeutic in binge-eating disorders, such as bulimia nervosa.

While there is an urgent need to identify inhibitors of ADAR2 that may have specificity for the RNA editing of 5-HT2CR, modifying the RNA editing of other ADAR targets may be beneficial. Epilepsy has been associated with increased RNA editing of glutamate receptors (kainate subtypes), the glutamate transporter EAAT2 (SLC1A2), CTSB, Rpa, Sparc, and OVCA2 (Figure 1). Therefore, inhibiting the RNA editing activity of ADAR2 may be beneficial for the treatment of some forms of epilepsy. In addition, one transcriptome-wide study showed increased RNA editing of more than 100 mRNAs in the frontal cortex of schizophrenia patients. Therefore, ADAR2 inhibition may be therapeutic in schizophrenia and other psychotic disorders [6].

Prion diseases have been associated with increased RNA editing of FKRP and Rragd in mice, and may be improved by administration of ADAR inhibitor drugs. Moreover, elevated RNA editing activity has been reported as a major contributor to transcriptomic diversity in tumors [7], therefore an inhibitor of RNA editing may be therapeutic in some forms of cancer. ADAR enzymes modify the sequence of double-stranded RNA loop structures, which includes the nucleic acids within RNA viruses, often with proviral effects (Figure 1). Effects of editing have been documented only in a few groups of viruses, such as HIV, measles, Zika virus, SARS-CoV-2, mink enteritis virus (MEV), Ebola virus (EBOV), human parainfluenza viruses (HPIV), mumps virus (MuV), vesicular stomatitis virus (VSV), plant viruses and Sendai virus (SeV) [26]. Inhibition of ADAR enzymes may be therapeutic in the treatment of infections caused by some RNA viruses.

The availability of NMR and X-ray structures of the human ADAR2 (hADAR2) with the dsRNA of GluA2 offer an opportunity for the in silico screening of potential inhibitors of RNA editing [27,28]. This project therefore sought to employ structure-based drug design (SBDD) methods to identify natural products-derived compounds with potential inhibitory activity against the hADAR2 enzyme. SBDD approaches have become beneficial in identifying small molecules for treating various diseases [29] and natural products serve as an extensive reservoir of therapeutic candidates [30]. Natural products are structurally diverse and can be explored for the treatment of various disorders [31,32]. Since antiquity, traditional Chinese medicine has been used to improve neural regeneration and to repair neurological disorders [33,34,35]. Therefore, this study carefully predicted good binders of the ADAR2 with probable inhibitory activity by using structure-based virtual screening, molecular dynamics simulations and molecular mechanics Poisson–Boltzmann surface area calculations using compounds obtained from the Chinese flora and fauna. The study further predicted the biological activity and toxicity profiles of the identified biomolecules.

## 2. Results and Discussions

### 2.1. Prediction of Binding Sites

ADAR proteins share a common structure which consists of N-terminal dsRNA binding domains (dsRBDs) and a carboxy (C)-terminal catalytic deaminase domain [36,37]. The human ADAR2 (hADAR2) has been reported to contain two dsRBMs, which are separated by a 90-amino acid linker, and followed by the C-terminal catalytic domain [37]. The N terminus has been suggested to contain sequences that are involved in the auto-inhibition of the enzyme [37]. The IHP binds in the active site of the hADAR2, which is lined by residues Asn391, Ile397, Arg400, Arg401, Thr513, Lys519, Arg522, Gly530, Ser531, Lys629, Tyr658, Lys662, Tyr668, Lys672, Glu689, Lys690 and Asp695 [27,38]. A previous study has suggested that ADAR2 activation requires binding to an RNA which interacts with both double-stranded RNA binding motifs (dsRBMs) of the ADAR2 [37]. The IHP has also been suggested to activate ADAR2 upon binding [27,38].

Computed Atlas of Surface Topography of proteins (CASTp) predicted 52 potential binding sites, out of which only 3 were selected as plausible based on the area and volumes of the cavities (Figure 2 and Table 1). The other predicted sites either had no openings or were relatively small, such that no ligand could dock into [39,40]. Pocket 1 as predicted via CASTp (Figure 2 and Table 1) is consistent with the RNA binding loop of the ADAR2 [27,41], while pocket 2 is the IHP-binding site of the ADAR2 [27,38] (Figure 2 and Table 1).

### 2.2. Molecular Docking

#### 2.2.1. Validation of Docking Protocols

The classical root mean square deviation (RMSD) was used to assess the ability of each docking tool to predict poses similar to that of the crystallographic structure. RMSD equal to or lower than 2 Å is considered good, 2 Å < RMSD < 3 Å is acceptable and RMSD > 3 Å is bad [43]. However, RMSD < 2 Å cut-off value is widely regarded as the most effective threshold value for validating correctly posed molecules [44,45]. The most negative IHP poses from AutoDock Vina and Glide had RMSD values of 1.6516 and 0.893 Å, respectively, when compared to the crystallographic or reference structure. The RMSD values obtained herein indicate the near accuracy pose prediction of AutoDock Vina and Glide for ADAR2 binders.

#### 2.2.2. Molecular Docking via AutoDock Vina

A total of 25,131 compounds were successfully screened against the human ADAR2 protein. For each compound, AutoDock Vina generates up to nine conformers during docking. The pose or conformation with the most negative binding energy was selected as the best for each compound. The top 10% of the initial docking library (that is 2520 compounds) were shortlisted for analysis. The bound ligand, IHP had a binding energy of −8.6 kcal/mol when docked against the hADAR2. For the TCM natural products, ZINC000095913861 had the most negative binding energy when docked against the ADAR2 protein with a binding energy of −12.2 kcal/mol. ZINC000044350981 had a binding energy of −11.4 kcal/mol, while ZINC000070450936 and ZINC000070454365 both had a binding energy of −11.2 kcal/mol. Compounds ZINC000003203078, ZINC000085546044, ZINC000085593200, ZINC000085594057, and ZINC000103536976 had a binding energy of −11.1 kcal/mol.

The highest binding energy was observed to be −8.9 kcal/mol among the top 10%, which is lower than the −7.0 kcal/mol threshold specified for AutoDock Vina to differentiate between putative binders and non-binders of proteins [46]. The −7.0 threshold has been shown to filter ~95% of non-inhibitors while passing ~98% of known inhibitors [46]. The more negative binding energies of the shortlisted compounds (−8.9 kcal/mol and lesser) suggest that a greater number of the predicted ligands may be good ADAR2 binders. Additionally, the IHP had the least binding affinity (highest binding energy) to the hADAR2 as compared to the top 10%.

#### 2.2.3. Molecular Docking via Maestro

All 37,398 TCM compounds generated via LigPrep were screened against the ADAR2 protein using Glide [47]. For compounds with two or more tautomers, the pose with the most negative binding energy was chosen. The control, IHP, had a binding energy of −7.97 kcal/mol. ZINC000103569281 had the most negative binding energy of −9.892 kcal/mol, followed by ZINC000013432666, ZINC000085488553, and ZINC000103584108 with docking scores of −9.718, −9.606, and −9.546 kcal/mol, respectively. Among the top 10%, the compounds which had the highest binding energies (least affinity) with the ADAR2 protein had docking scores of −6.10 kcal/mol.

Most of the top compounds were observed to have indole, naphthoquinone, coumarin, furanocoumarin or benzofuran moieties in their structures. Compounds including ZINC000030726422, ZINC000070450892, ZINC000070454387, ZINC000070455595, and ZINC000085492822, among others were observed to have the indole moiety. Serotonin or 5-hydroxytryptamine (5-HT) and tryptophan also have the indole moiety. Tryptophan is the only precursor of serotonin production in humans [48]. Tryptophan has been shown to regulate mood and cognition [49]. Serotonin and tryptophan are known to contribute to well-being and happiness [50,51,52,53]. Serotonin reduces anxiety and promotes good mood and happiness by regulating brain function and cognition [54]. Serotonin also helps regulate the precursor to melatonin, a major control hormone of the sleep cycle [55,56,57,58] and digestive system regulation [49]. Selective serotonin reuptake inhibitors (SSRIs), including sertraline and fluoxetine, have been shown to have positive effects on patients with neurological or neurodevelopmental disorders including autism spectrum disorders, depression, anxiety, and obsessive-compulsive disorder [59,60,61,62].

Compounds ZINC000014690026, ZINC000070455383, ZINC000085488553, ZINC000085488571, and ZINC000085488602, among others had the benzofuran and/or coumarin moieties in their structure. Coumarins have recently been explored for treating neurological or brain disorders [63,64,65,66,67]. These chemical series need further exploration in order to identify more potent and effective ADAR2 inhibitors to increase our therapeutic arsenal against ADAR2-implicated disorders. De novo design and lead optimization via scaffold hopping, bioisosteric replacement, and other available methods will aid in identifying more effective and potent anti-ADAR2 compounds from these chemical series [68,69,70,71]. New pharmacologic agents from these series may demonstrate novel mechanisms of action with improved efficacy, tolerability and safety.

#### 2.2.4. Shortlisting Compounds Based on Consensus Score

A total of 516 compounds were observed to be in the top 10% for both AutoDock Vina and Maestro. A consensus docking score was calculated by determining the averages of the AutoDock Vina binding energy and the Glide docking score of the top compounds. The 516 compounds were then ranked based on the consensus score. The known ADAR2 binder, IHP, had a consensus score of –8.26 kcal/mol (Table 2). ZINC000049888963 had the most negative consensus score of –9.58 kcal/mol followed by ZINC000044417732 (Table 2), ZINC000095909990, ZINC000095909350, and ZINC000085509666 with scores of –9.42, –9.41, –9.34, and –9.27 kcal/mol, respectively. Compounds ZINC000085505346, ZINC000085950180 (Table 2), and ZINC000095909587 also had consensus scores of –9.24, –9.19, and –9.13 kcal/mol, respectively. ZINC000085569204 had the least negative or highest consensus score of –7.50 kcal/mol among the 516 shortlisted compounds.

The study analyzed the effect of molecular weight on the binding scores. No compound with molecular weights 150 g/mol to 250 g/mol were found in the top 516 hits (Figure 3). It was also observed that the compounds with molecular weights between 250 g/mol and 350 g/mol had lower binding affinity (higher binding energy) than those with molecular weight above 350 g/mol (Figure 3 and Appendix A). All compounds with molecular weight <350 g/mol had consensus docking scores higher than −8.5 kcal/mol (Figure 3). Similar trends were observed for the AutoDock Vina and the Glide binding energy scores of the top 516 compounds. For AutoDock Vina and Glide, no compound with molecular weight <350 g/mol had binding energies lower than −10.4 (Appendix A) and −7.7 kcal/mol (Appendix A), respectively.

### 2.3. ADAR2-Ligand Interaction Profiling

The interaction profiles of the hADAR2 protein–ligand complexes for the shortlisted compounds were determined using Maestro. The best docking conformation from the Glide docking results were used to study the ADAR2-ligand interactions. IHP was predicted to form 12 hydrogen bonds with residues Asn391, Arg400, Arg401 (2 H-bonds), Lys519 (2 H-bonds), Ser531, Lys672, Trp687, Val688, Glu689, and Lys690 of the ADAR2 protein (Figure 4A and Table 2). It also formed salt bridges with Arg401, Lys519, Arg522, Lys629, Lys662, Lys672, and Lys690 (Figure 4A and Table 2). ZINC000049888963, which had the most negative consensus docking score, interacted via 7 H-bonds with the ADAR2 including residues Arg400, Lys519, Trp523, Gln529, Leu532, and Lys690 (2 H-bonds). It was also involved in 4 pi-cation interactions with Arg400, Arg522, and Lys629 (2 contacts). ZINC000044417732 formed hydrogen bonds with Arg401, Leu532, Lys662, and Glu689; pi-cation interactions with Arg400 and Lys662; and salt bridges with Arg400, Arg401, Lys629, and Lys662 (Figure 4B and Table 2). 

ZINC000095909990 formed 8 hydrogen bonds with residues Asn391, Tyr408, Met514, Arg522, Lys629, Lys662, Val688, and Pro691. It also formed pi-cation interactions with Lys629 and Lys662, and four salt bridges with Arg400, Lys519 (2), and Lys690. ZINC000095909350 had four hydrogen bond interactions with residues Asp392, Ser531, His659, and Val688; five pi-cation interactions with Arg400, Arg401, Lys519, Lys662, and Lys672; and five salt bridges with residues Arg400, Arg401, Lys519, Lys629, and Lys662. ZINC000085509666 interacted with ADAR2 via 10 hydrogen bonds with residues Asn391, Leu532, Gln529, Lys629, Tyr658, Lys662, Glu689 (3 H-bonds), and Lys690; 2 salt bridges with Lys519 and Lys690; and formed a pi-cation with Arg400.

ZINC000085505346 interacted with ADAR2 via hydrogen bonds with Arg400, Arg401 (2), Trp523, Gln529, Leu532, Lys629, His659, and Trp687; pi-cation interaction with Arg522; and salt bridges with Arg522, Lys672, and Glu689. Compound ZINC000085950180 formed six hydrogen bonds with Arg401, Ser531 (2 H-bonds), Lys629, Trp687, and Asp695 and a pi-cation interaction with Lys662 (Table 2). ZINC000095909587 was involved in 6 hydrogen bonds with residues Arg522, Ser531, Lys629, Tyr668, Trp687, and Val688; and formed 2 salt bridges with Lys519 and Lys690. Multiple hydrogen bonds existing in protein–ligand complexes influence ligand binding affinity and contributes to ligand activity [72,73]. Residues Arg400, Arg401, Lys519, Ser531, Leu532, Trp687, and Lys690 were observed to interact with most of the ligands and may be crucial for ligand binding. The protein–ligand interaction profiles of the potential lead compounds are shown in Figure 4B and Appendix AA–H.

### 2.4. ADMET Prediction

The top 516 compounds were subjected to ADMET testing via SwissADME to predict their pharmacokinetic profiles [74]. The compounds were assessed using Lipinski’s and Veber’s rules [75,76,77,78,79]. Lipinski’s rule of five requires an orally active drug to have less than two violations of four criteria: less than 5 hydrogen bond donors; less than 10 hydrogen bond acceptors; molecular mass less than 500 Da; and an octanol-water partition coefficient (logP) less than 5 [75,79]. Veber’s rule, on the other hand, requires compounds with good oral bioavailability to have not more than 10 rotatable bonds and topological polar surface area (TPSA) not more than 140 Å^2^ [78]. Fluoxetine (antidepressant), nebularine (ADAR inhibitor) and doxorubicin (anti-cancer) were used as controls for the ADMET prediction. Fluoxetine and nebularine were predicted to not violate both Lipinski’s and Veber’s rules, while doxorubicin had 3 and 1 Lipinski’s and Veber’s violations, respectively (Table 3). 

Compounds which violated two or more components of Lipinksi’s rule of five and/or Veber’s rule were eliminated. A total of 343 and 447 compounds failed Lipinski’s and Veber’s rules, respectively. Compound ZINC000049888963, which had the most negative consensus docking score, failed both Lipinski’s and Veber’s rules with 3 and 1 violations, respectively. ZINC000049888963 had a TPSA of 216.58 Å^2^, molecular weight of 594.52 g/mol, 13 hydrogen bond acceptors and 7 hydrogen bond donors. ZINC000044417732, the second most negative compound with consensus score of –9.42 kcal/mol, passed both Lipinski’s and Veber’s rules, having a TPSA of 108.74 Å^2^; 6 and 2 H-bond acceptors and donors, respectively; and a molecular weight of 374.34 g/mol. In total, 69 compounds passed both Lipinski’s and Veber’s rules and were thus considered for further analysis. The molecular weight of the 69 compounds ranged between 262 and 599 g/mol, while their TPSAs ranged between 58 and 140 Å^2^. 

Most neuroactive and psychoactive drugs bypass the blood-brain barrier (BBB) to be effective [80]. Thus the BBB permeability of the shortlisted compounds and controls were predicted. Fluoxetine was predicted as BBB permeant while nebularine and doxorubicin were predicted as non-permeant (Table 3). Only four compounds, comprising ZINC000013462928, ZINC000014612330, ZINC000100014196, and ZINC000100513617 were predicted to permeate the blood-brain barrier, making these compound interesting candidates to probe further. The BBB is a huge challenge for the development of neuroprotective drugs [81]. However, studies have shown that there are other effective ways of administering drugs bypassing the BBB [81,82,83,84]. The intranasal route, which is a noninvasive delivery route to bypass the barrier, can be exploited to administer compounds which cannot permeate the BBB [81,82]. Additionally, direct brain administration, which is an invasive approach to target the brain region with therapeutic molecules, can also be considered [82].

OSIRIS Datawarrior version 5.5.0 was also employed to predict toxicity properties including mutagenicity, tumorigenicity, irritancy and reproductive effects of the 69 shortlisted compounds [85] (Appendix A). Datawarrior predicts the toxicity risk of compounds by classifying them as “none”, “low” or “high” for each of the four toxicity properties. Compounds that were predicted to possess low or high toxicity for more than one property were eliminated. Moreover, compounds that were predicted to be either mutagenic or tumorigenic were also removed from the potential hit list. Mutagens and tumorigenic agents were eliminated because ADAR2 inhibition by death associated with protein 3 (DAP3) has been reported to be involved in certain cancer type development and progression [86]. Down-regulation of ADAR2 in lung cancers has been previously reported [87]. ADAR2 is a tumor suppressor and its inhibition may promote tumor growth [88,89,90]. Thus, ADAR2 inhibitors with mutagenic or tumorigenic properties may promote or speed up neoplasm due to ADAR2 inhibition.

A total of 51, 64, 52 and 52 compounds were predicted to be non-mutagenic, non-tumorigenic, non-irritant and exhibiting no reproductive effects, respectively (Appendix A). However, only 29 compounds, including the top 3 comprising ZINC000044417732, ZINC000085950180 and ZINC000085511995, were predicted as non-mutagenic, non-tumorigenic, non-irritant and with no reproductive effects (Appendix A). Of the 4 compounds that were predicted to be BBB permeants, 3 comprising ZINC000014612330, ZINC000100014196, and ZINC000100513617 were predicted to be non-tumorigenic, non-mutagenic, non-irritant and had no reproductive effects, while ZINC000013462928 was predicted to pose no toxicity risk except for its high reproductive effect (Appendix A).

A total of 13 compounds were predicted to have two or more toxicity risks and were thus eliminated from the study. Moreover, a total of 9 compounds comprising ZINC000085569519, ZINC000085569501, ZINC000003918875, ZINC000085569417, ZINC000085569484, ZINC000085594490, ZINC000085569502, ZINC000085569474, and ZINC000059588402 were predicted to have high mutagenic effects while 9 others posed low mutagenic risks. The 9 compounds with low mutagenic risks include ZINC000014613520, ZINC000014814624, ZINC000004098700, ZINC000095912516, ZINC000015214955, ZINC000015254000, ZINC000095911983, ZINC000059587863, and ZINC000005714910. Furthermore, two (ZINC000059588402 and ZINC000085569204) and three compounds (ZINC000085594490, ZINC000103543244, and ZINC000059587863) were predicted to have high and low tumorigenic effects, respectively, and were also eliminated from the study (Appendix A). In all, 47 compounds passed the toxicity test and were considered as potential lead compounds with insignificant safety and toxicity concerns (Appendix A).

### 2.5. Structural Similarity Search and Prediction of Biological Activity of Shortlisted Compounds

The biological activity of the identified compounds were predicted using the Prediction of Activity Spectra of Substances (PASS) by submitting the SMILES format of the compounds as inputs [91,92,93]. PASS predicts the activity of a molecule based on the similarity of the query with the structures of molecules available in its training dataset. The probability “to be active” (Pa) provides an estimation for the likelihood of the query compound to belong to sub-set of active compounds, whereas the probability “to be inactive” (Pi) provides an estimation for the query to be similar to the inactive subset of PASS training dataset. Compounds whose activities have Pa > Pi are considered likely to exhibit those activities.

Four known drugs namely quercetin, nebularine, lovastatin, and simvastatin were used as controls. Quercetin and nebularine, which are known adenosine deaminase (ADA) inhibitors [94,95,96,97,98], were used as positive controls while lovastatin and simvastatin were the negative controls. Lovastatin and simvastatin are both 3-hydroxy-3-methylglutaryl-CoA (HMG-CoA) reductase inhibitors which suppress cholesterol synthesis (IC_50_ < 5 µM) [99]. Both compounds did not inhibit ADA nor increase adenosine in hepatocytes, thus having no effect on autophagy [99]. Quercetin on the other hand demonstrated ADA inhibition with an IC_50_ of 170 µM (0.17 mM) in rats [94] while nebularine has been reported to inhibit ADA with an IC_50_ of 90 µM [100]. Nebularine is also a known ADAR inhibitor [98]. Both lovastatin and simvastatin showed no deaminase inhibition properties by the PASS predictions, supporting previous study that they may not be ADA inhibitors [99].

On the other hand, nebularine was predicted by PASS to be an inhibitor of AMP deaminase 2 and 3, and 11 other deaminases including adenine (Pa: 0.942 and Pi: 0.000), adenosine (Pa: 0.619 and Pi: 0.002), glucosamine-6-phosphate (Pa: 0.519 and Pi: 0.006), cytosine (Pa: 0.486 and Pi: 0.001), guanine (Pa: 0.404 and Pi: 0.003), deoxycytidine (Pa: 0.350 and Pi: 0.002), cytidine (Pa: 0.298 and Pi: 0.008), dCTP (Pa: 0.275 and Pi: 0.001), dCMP (Pa: 0.193 and Pi: 0.001), blasticidin-S (Pa: 0.205 and Pi: 0.066), and AMP (Pa: 0.118 and Pi: 0.001) deaminases. Nebularine was also predicted to be antineoplastic (Pa: 0.702 and Pi: 0.026) and antiviral (Pa: 0.886 Pi: 0.002). Quercetin was also predicted to be inhibitors of 8 deaminases including pterin (Pa: 0.352 and Pi: 0.104), blasticidin-S (Pa: 0.213 and Pi: 0.059), creatinine (Pa: 0.184 and Pi: 0.067), adenosine (Pa: 0.103 and Pi: 0.014), ornithine cyclodeaminase (Pa: 0.165 and Pi: 0.113), cytidine (Pa: 0.101 and Pi: 0.079), deoxycytidine (Pa: 0.093 and Pi: 0.070), and glucosamine-6-phosphate (Pa: 0.105 and Pi: 0.083) deaminases. Quercetin was also predicted as antineoplastic (Pa: 0.797 and Pi: 0.012) and antiviral (Pa: 0.498 and Pi: 0.005). The relatively higher Pa values (and lower Pi) for nebularine supports nebularine’s better ADA inhibition than quercetin. 

ZINC000044417732, ZINC000085950180, and ZINC000085511995 were predicted as inhibitors of various deaminases including ornithine cyclodeaminase, pterin, creatinine, glucosamine-6-phosphate, cytosine, deoxycytidine and ATP deaminases. The ADAR2 also belongs to the deaminase class of proteins, implying the potential of these compounds to inhibit the ADAR2. Furthermore, ZINC000014612330, ZINC000044417732, ZINC000085950180, and ZINC000085511995 were predicted to be useful in the treatment of acute neurologic disorders with Pa values of 0.627, 0.562, 0.562, and 0.442; and Pi values of 0.031, 0.051, 0.051, and 0.116, respectively. ZINC000085511995 and ZINC000100513617 were further predicted as antineurotic. Some of the compounds were predicted as neurotransmitter uptake inhibitors. Neurotransmitter uptake inhibitors, such as the SSRI class of drugs including sertraline and fluoxetine, have positive effects on depression, anxiety disorders, and certain types of obsessive-compulsive disorder [59,60,61]. Sertraline and fluoxetine are also beneficial for treating depression in epilepsy patients since they also lower the risk of triggering seizures [101]. 

ZINC000085511995 was predicted as an antidyskinetic, which might be relevant in treating dyskinesia in most neurological disorders. Most of the compounds were also predicted to be good for movement disorder treatment. Most neurological disorders including schizophrenia and autism spectrum disorder, are characterized by movement disorders [102,103,104,105,106,107]. The compounds were also predicted as kidney function stimulants with Pa values of 0.524, 0.526, and 0.711 and Pi values of 0.082, 0.080, and 0.008, respectively. Various studies have associated schizophrenia with high chronic kidney disease (CKD) risk [108,109], therefore these compounds may be helpful in treating renal complications associated with these disorders. 

All the potential lead compounds, except ZINC000014612330 and ZINC000013462928, were predicted as inhibitors of various enzymes involved in phosphoinositide processes including inositol oxygenase, glycosylphosphatidylinositol phospholipase D, CDP-diacylglycerol-inositol 3-phosphatidyltransferase, glycosylphosphatidylinositol diacylglycerol-lyase, guanidinodeoxy-scyllo-inositol-4-phosphatase, inositol-3-phosphate synthase, phosphatidylinositol diacylglycerol-lyase and inositol 1,4,5-triphosphate 3-kinase. Since IHP is required for ADAR2 activation [27,38], inhibiting its production and transfer may help limit ADAR2 activity. ZINC000044417732, ZINC000085950180, and ZINC000085511995 were predicted as apoptosis agonists with Pa values of 0.837, 0.623 and 0.707; and Pi values of 0.005, 0.024 and 0.014, respectively. DAP3, an ADAR2 inhibitor has been previously reported to be involved in mitochondrial physiology and cell apoptosis [110,111]. Moreover, reduced expression of ADAR1 and ADAR2 have been reported to increase apoptosis in a region-specific manner in the hippocampus, parietal cortex and sub-ventricular zone of seizure-exposed brains in mouse [112].

Moreover, ZINC000044417732, ZINC000085950180, and ZINC000085511995 were predicted as Pin1 inhibitors with Pa values of 0.699, 0.702, and 0.807; and Pi values of 0.007, 0.007, and 0.004, respectively. Previous studies have shown that ADAR2-Pin1 interactions stabilizes ADAR2 in the nucleus [113,114]. Additionally, nuclear interactions between Pin1 and ADAR2 were observed to increase as neurons develop or mature [114]. Pin1 is also required for the editing of expressed GluA2 transcripts in cell lines [113] and to regulate ADAR2 protein levels as well as catalytic activity [113]. Pin1 is also known to facilitate multiple cancer-driving processes [115], supporting the prediction of these compounds as potential anti-cancer molecules. Aberrant A-to-I RNA editing has been shown to be implicated in cancers [86,116]. DAP3, which is an ADAR2 inhibitor, has been reported to affect editing targets involved in cancer-related signaling pathways and processes [86]. However, ZINC000044417732, ZINC000085950180, and ZINC000085511995 were predicted to be antineoplastics with Pa values of 0.839, 0.563 and 0.805; and Pi values of 0.008, 0.053 and 0.011, respectively. Furthermore, ZINC000044417732, ZINC000085950180, ZINC000085511995, and ZINC000085850673 were predicted as antimutagenic with Pa values of 0.793, 0.777, 0.845, and 0.795; and Pi of 0.004, 0.004, 0.003, and 0.004, respectively. They were also predicted to be beneficial in treating preneoplastic conditions.

ZINC000085511995 (mamegakinone) was also predicted to be similar to vitamin K with a Pa of 0.39 and a Pi of 0.002. Further structural similarity search via DrugBank revealed that ZINC000085950180 (isozeylanone) is similar to phylloquinone (vitamin K1), menatetrenone (vitamin K2), menaquinone-7, menaquinone-6 and menaquinone with similarity scores of 0.75, 0.748, 0.748, 0.748, and 0.748, respectively. ZINC000085734971 was also predicted to be structurally similar to these compounds. All the aforementioned compounds are naphthoquinones, a class of compounds widely known for their anti-cancer properties [117,118,119,120]. Naphthoquinones have demonstrated remarkable applications in medicinal chemistry owing to their good synthetic accessibility, making it possible to obtain many chemical substances [121]. Compounds ZINC000085950180 and ZINC000085511995 are composed of two naphthoquinones each in their structures. High levels of vitamin K has been shown to improve cognitive function and its deficiency may be associated with an increased risk of cognitive decline and dementia in older adults [122,123].

ZINC000014612330 is structurally similar to lapachone with a score of 0.636. β-lapachone has been reported to attenuate cognitive impairment and neuroinflammation in an Alzheimer’s disease mouse model [124]. ZINC000034517814 is structurally similar to acteoside and echinacoside, both with a similarity score of 0.752. Both compounds have been shown to be beneficial in the treatment of neurological conditions including Parkinson’s and Alzheimer’s diseases [125,126,127]. Echinacoside has been shown to selectively reverse dopaminergic neuronal injury in rat models of Parkinson’s disease induced by rotenone [126]. However, echinacoside and acteoside have significant pharmacokinetics concerns as they have very fast absorption and elimination rates in rats [128]. In humans, echinacoside was not identified in plasma samples after echinacea tablets were ingested [129]. Herein, SwissADME predicted a bioavailability score of 0.17 for both echinacoside and acteoside, which corroborates the poor bioavailability of both compounds [128,129]. However, ZINC000034517814 had a better score of 0.55, suggesting that ZINC000034517814 has the best oral bioavailability compared to echinacoside and acteoside [130].

ZINC000100014196 (indigo), ZINC000100513617 (indigo) and indirubin are isomers [131,132]. About 2.3 to 23.3 µg/kg of an indirubin derivative, and 7-bromoindirubin-3-oxime, has been recently shown to prevent β-amyloid (Aβ) oligomer-induced impairments of spatial cognition and recognition in mice and has been suggested for the treatment of Alzheimer’s disease [131]. Its anti-cancer activity at an IC_50_s less than 35 μM has also been reported [132,133]. ZINC000013462928 is structurally similar to arctigenin and matairesinol with scores of 0.632 and 0.626, respectively. Both arctigenin and matairesinol have been reported to have neuroprotective, anti-diabetic and anti-depressive activities [134,135,136]. The predicted biological activities of the shortlisted compounds corroborate their potential anti-ADAR2 activity.

### 2.6. Molecular Dynamics Simulations

This study investigated the structural conformation changes and atomic motions by performing 100 ns MD simulations on the unbound protein and ten ADAR2-ligand complexes. The protein–ligand complexes investigated include ADAR2 bound with IHP, ZINC000044417732, ZINC000085950180, ZINC000085511995, ZINC000085850673, ZINC000085996580, ZINC000085734971, ZINC000014612330, ZINC000100513617, and ZINC000013462928. The top six compounds based on the most negative consensus docking score and good toxicity profiles and three compounds that were predicted as BBB permeants (ZINC000085734971, ZINC000014612330, and ZINC000100513617) were selected for the MD simulations. ZINC000100014196 (indigo) and ZINC000100513617 are isomers and thus the latter was selected for the MD simulation due to its higher binding affinity. The structural stability, folding and conformational fluctuations of the proteins caused by ligand binding were investigated using the RMSD, radius of gyration (Rg) and the root mean square fluctuation (RMSF) after the 100 ns simulations. Hydrogen bond analysis throughout the 100 ns simulation period were performed for each complex. Snapshots were also generated to assess the position of the ligands with respect to the ADAR2 protein.

#### 2.6.1. RMSD of ADAR2 and ADAR2-Ligand Complexes

The RMSD measures the deviation of the final conformation of the protein’s backbone with regards to the protein’s initial structural conformation [137]. RMSD provides insights into the stability of a protein—a protein with lower RMSD tends to be more stable than a protein with higher deviations in its backbone [137,138]. Herein, all structures reached equilibrium after ~15 ns (Figure 5 and Appendix A). The unbound protein experienced minor fluctuations until about 50 ns and then remained stable till the end of the 100 ns simulation period with an average RMSD of 0.23 nm (Figure 5). The ADAR2-IHP (Figure 5) and the ADAR2-ZINC000014612330 (Appendix A) complexes were observed to be more stable than the unbound protein throughout the simulation period, maintaining RMSD averages of 0.21 and 0.22 nm, respectively. Although the ADAR2-IHP complex showed the greatest stability among all the structures, the ADAR2-ZINC000014612330 complex demonstrated comparable stability (Appendix A). The stability of the ADAR2-IHP is not surprising since the IHP is a known binder of ADAR2 and is required for ADAR2 activation [27,38]. 

The ADAR2-ZINC000085511995 complex was mostly stable throughout the simulation period. It reached relative equilibrium at around 10 ns (0.225 nm), maintained stability till 75 ns where the RMSD rose to ~0.3 ns and fell to 0.25 nm (around 80 ns) and remained stable till the end (Figure 5). The ADAR2-ZINC000100513617, ADAR-ZINC000085734971, and ADAR2-ZINC000085850673 complexes experienced similar RMSD trends (Figure 5 and Appendix A). The ADAR2-ZINC000100513617 complex rose to 0.227 nm at around 50 ns and remained stable till the end of the simulation time with an average RMSD of 0.23 nm (Appendix A). The RMSD of ADAR2-ZINC000085850673 complex rose to 0.25 nm at around 35 ns and maintained stability till the end of the simulation with an average of 0.25 nm (Figure 5). The ADAR2-ZINC00008573491 complex rose to 0.25 nm at around 25 ns and remained stable till the end of the 100 ns simulation period (Appendix A).

The ADAR2-ZINC000085950180, ADAR2-ZINC000013462928, and ADAR2-ZINC000085996580 complexes were the least stable among all the structures. The ADAR2-ZINC000085950180 complex rose to 0.2 nm at around 25 ns, then rose to 0.25 nm at 50 ns till about 80 ns and then remained stable till the end of the 100 ns period with an average RMSD of 0.24 nm (Figure 5). For the ADAR2-ZINC000085996580 complex, the RMSD experienced a gradual rise till about 35 ns, maintained an average of 0.26 nm till about 70 ns, then rose to about 0.3 nm (Figure 5). The RMSD of the ADAR2-ZINC000013462928 complex rose to 0.25 nm after 17 ns and remained stable till about 70 ns, where it experienced a rise to 0.3 nm (around 80 ns) and maintained it till 100 ns (Appendix A).

#### 2.6.2. Radius of Gyration of ADAR2 and ADAR2-Ligand Complexes

The radius of gyration provides information about the stability, folding and compactness of a protein, and can be further defined as the RMSD of atoms from the centroid of a protein [139,140]. The Rg plots of all the structures (unbound protein and complexes) revealed good stability and compactness of the ADAR2 protein. The Rg values of all the systems ranged between 2.04 nm and 2.1 nm (Figure 6 and Appendix A), signifying their stable folding [140]. The unbound protein maintained an average Rg of about 2.085 nm till about 27 ns, then fell to 2.055 nm at around 40 ns, then it rose and maintained an average of 2.07 ns until the end of the simulation (Figure 6). The IHP maintained an average Rg of 2.06 nm throughout the simulation (Figure 6). The ADAR2-ZINC000044417732 complex was relatively stable, maintaining an average of 2.075 nm throughout the simulation, although a high rise in Rg (to about 2.095 nm) was observed between 50 and 60 ns (Figure 6). The ADAR2-ZINC000085950180, ADAR2-ZINC000014612330 and ADAR2-ZINC000085850673 complexes demonstrated similar trends (Figure 6 and Appendix A). All three complexes were very stable throughout the simulation period, maintaining an average of 2.065 nm. The Rg of the ADAR2-ZINC000085734971 complex was also very stable until 80 ns with an average of ~2.05 nm, where a rise to 2.06 nm was observed (Appendix A). 

The Rg values of ADAR2-ZINC000085996580 complex fell until ~50 ns where a sharp rise was observed (Figure 6). The Rg rose to ~2.09 averagely until 90 ns and then fell again (Figure 6). The ADAR2-ZINC000100513617, ZINC000013462928, and ZINC000085511995 complexes also demonstrated fluctuations similar to that of the ADAR2-ZINC000085996580 complex. The ADAR2-ZINC000013462928 complex maintained quite a stable Rg with an average of 2.075 nm until ~80 ns and then declined to an average of 2.06 nm until the end of the 100 ns simulation time (Appendix A). The Rg of the ADAR2-ZINC000100513617 complex, however, increased after 90 ns (Appendix A). Collectively, all the complexes and the unbound protein had relatively low Rg values, indicating that they were stable and compact in their folding.

#### 2.6.3. RMSF of ADAR2-Ligand Complexes

Herein, the RMSF of the various complexes were evaluated to determine the ADAR2 residues involved in the different mobility in the RMSD plots [139]. The RMSF in the binding cavity provides insights into the residues that make strong interactions with the ligand [139]. A high RMSF indicates the residues that do not make strong interaction, and thus have higher mobility [139].

For all the complexes, fluctuations were observed at similar residue indexes. Major fluctuations were observed at residue indexes 460–480, 490–515, and 582–600 (Appendix A). Minor fluctuations were observed at residues 340–350, 360–370, 380–390, 415–430, 565–578, 620–630, and 650–665 (Appendix A). ZINC000085996580 induced the highest fluctuations on the ADAR2 protein at residues Ala468 and Gln507 with RMSF values of 0.7276 and 0.7262 nm, respectively. For the ADAR2-ZINC000085996580 complex, residues Asp469, Arg470, and Leu506 were also observed to have high fluctuations with values of 0.6178, 0.647, and 0.6638 nm, respectively. 

Residue indexes 390–415, 440–450, 515–560, 595–620, 635–650, and 660–700 experienced the least fluctuations, which is suggestive of their very strong interactions with the ligands in the active site [139]. For residue indexes 390–415, the interactions with Arg400 and Arg401 could be the reason for the low RMSF values, while Lys519, Ser531 and Leu532 account for the low fluctuations around the indexes 515–560. The interactions with residues Trp687, Glu689 and Lys690 account for the low RMSF observed in the residues 660–700 region.

#### 2.6.4. Snapshots and Hydrogen Bond Analysis of Complexes

Since RMSD of the complex is calculated based on the protein backbone, snapshots were generated to verify the position of the ligands during the simulation [141]. Snapshots of each complex during the 100 ns MD simulation were generated at 25 ns intervals. For all the complexes, the ligands were tightly bound to the IHP binding site of the ADAR2. Additionally, aligning the resulting structures with the initial structure (time = 0 ns) as reference, further corroborated the stability of the complexes throughout the MD simulations. The RMSDs were determined for the structural alignment of each complex using the align module in PyMOL. 

The RMSD values between the structures at times 25, 50, 75, and 100 ns when aligned to the initial structure (time = 0 ns) were 1.064, 1.161, 1.074, and 1.231 Å, respectively, for the ADAR2-IHP complex. For the ADAR2-ZINC000044417732 complex, RMSDs of 1.052, 1.264, 1.395, and 1.631 Å, respectively, were observed. For the ADAR2-ZINC000085950180 complex, 1.091, 1.064, 0.945 and 1.050 Å, respectively, were the observed RMSDs. Moreover, the ADAR2-ZINC000085511995 complex had RMSDs of 0.914, 0.932, 1.108, and 1.272 Å, respectively. The ADAR2-ZINC000085850673 structures at 25, 50, 75 and 100 ns also had RMSD values of 0.991, 1.204, 1.196, and 1.189 Å, respectively, when aligned to the initial structure. The ADAR2-ZINC000085734971 complex had RMSD values of 1.082, 1.149, 1.001, and 1.104, Å, respectively. The ADAR2-ZINC000100513617 complex also demonstrated stability with RMSD values of 1.359, 0.988, 0.996 and 1.292 Å, respectively. The relatively low RMSDs demonstrated by these complexes corroborate their good stability observed in the RMSD plots (Figure 5 and Appendix A).

The ADAR2-ZINC000014612330 complex also had RMSDs of 1.332, 1.322, 1.393, and 1.355 Å, when the initial structure was compared to the structures at times 25, 50, 75, and 100 ns, respectively. Although there were minor deviations from the starting structure, the ADAR2-ZINC000014612330 complex remained very stable after 50 ns (Appendix A). Aligning the structures at 75 and 100 ns to the 50 ns structure revealed RMSDs of 0.890 and 0.816 Å, respectively, which are consistent with the RMSD plot (Appendix A), implying that the ADAR2-ZINC000014612330 complex remained stable after 50 ns.

For the ADAR2-ZINC000085996580 complex, RMSDs of 1.414, 1.420, 1.330, and 1.360 Å were obtained when the structures at times 25, 50, 75, and 100 ns, respectively, were compared to the starting structure. Moreover, the ADAR2-ZINC000013462928 complex at times 25, 50, 75, and 100 ns demonstrated RMSDs of 1.291, 1.359, 1.415, and 1.233 Å, respectively, when aligned to the 0 ns structure. The relatively high RMSDs observed for ADAR2-ZINC000085996580 and ADAR2-ZINC000013462928 complexes (as compared to the other complexes herein) are consistent with the RMSD plot (Figure 5 and Appendix A) where the least stability (most deviation) was seen.

This study also analyzed the number of hydrogen bond interactions between the protein and each ligand during the MD simulations. Only hydrogen bond interactions within 0.35 nm and with an angle of 30° were reported using the GROMACS “gmx hbond” analysis (Figure 7). Snapshots of the complexes at 0 and 100 ns were visualized using Maestro and protein–ligand interaction profiles were also determined. The predicted number of hydrogen bonds via “gmx hbond” may be different from what is predicted via a molecular interaction visualization tool due to the cut-off.

For the ADAR2-IHP complex, 10 hydrogen bonds were within the 0.35 nm distance and 30° angle threshold as predicted via GROMACS although 12 were identified from the protein–ligand interaction profile. The 12 hydrogen bonds were formed with residues Arg400 (2 H-bonds), Arg401 (2 H-bonds), Lys519 (2 H-bonds), Arg522, Tyr658, Trp687, Val688, Glu689, and Lys690. At 4 ns, the hydrogen bond interactions increased to 14, after which a decline in the number of hydrogen bonds was observed. At the end of the 100 ns simulation, 8 hydrogen bonds with Thr513, Lys519, Tyr658, Lys662, Glu689 (2 H-bonds), Lys690, and Gln694 were observed.

For the ADAR2-ZINC000013462928 complex, 3 hydrogen bonds were predicted by GROMACS. However, only 2 hydrogen bonds with Met514 and Arg400 were seen using Maestro at 0 ns. At 100 ns, 3 hydrogen bonds with Tyr408, Ser531, and Leu532 were observed via Maestro while GROMACS predicted only one hydrogen bond. The hydrogen bond interactions with Arg400 and Met514 were lost after the 100 ns period. For the ADAR2-ZINC000014612330 complex, there was only one hydrogen bond with Ser531, prior to the MD simulation. After the 100 ns simulations, the hydrogen bond with Ser531 was lost and 3 new hydrogen bonds were formed with Arg401 and Trp523 (2 H-bonds) although only one passed the 0.35 nm and 30° angle cut-off.

At the start of the simulation, ADAR2-ZINC000044417732 had 4 hydrogen bonds with Arg401 and Lys662 (2 each) and at the end of the simulation, the complex maintained the 4 hydrogen bonds. However, the contacts with Lys662 were lost and 2 new hydrogen bonds with Asn391 and Tyr658 were formed. For ADAR2-ZINC000085511995, 4 hydrogen bond contacts with Arg401, Lys519, Arg522, and Trp687 existed before the MD simulation. After the simulation, only 3 were observed: 1 with Asn391 and 2 with Lys519. The ADAR2-ZINC000085734971 complex was predicted via “gmx hbond” to have 3 hydrogen bonds at both the initial and final states. However, after visualizing with Maestro, 4 hydrogen bonds with Ser531, Leu532, and Lys662 (2 bonds) were observed initially and 5 with residues Tyr408, Ser531 (2 bonds), Leu532, and Gln669 were observed at the end of the simulation. Compound ZINC000085850673 formed 3 hydrogen bonds with Met514, Lys519, and Tyr658 although 4 were predicted via GROMACS. After the simulation, 2 hydrogen bonds with Lys519 and Arg522 were observed. Hydrogen bond interactions with Met514 and Tyr658 were lost at the end of the simulation. 

The ADAR2-ZINC000085950180 complex had 3 hydrogen bonds with Ser531, Leu532, and Asp695 initially and 3 hydrogen bonds with Arg401, Tyr408, and Asp695 after the simulation (2 H-bonds were reported via “gmx hbond”). ZINC000085996580 was predicted to form 3 hydrogen bonds with ADAR2 at initial time, although only 1 bond with Ser532 was observed via Maestro. After the 100 ns MD simulation, 2 hydrogen bonds with Arg400 and Gln669 were observed. ZINC000100513617 formed 2 hydrogen bonds with Arg401 before the simulation (although 3 were predicted by “gmx hbond”) which were lost after the simulation. Multiple hydrogen bonds between a protein and a ligand have been shown to improve ligand binding and influence ligand activity [72,73]. However, the hydrogen bonds existing between both the donor and receptor must be either weaker or stronger than that between hydrogen and oxygen atoms in water, in order to influence ligand binding affinity [73]. Apart from ZINC000100513617, all the potential leads maintained hydrogen bonds with the ADAR2 throughout the simulation. 

Residues Arg400, Arg401, Lys519, Ser531, and Leu532 were observed to be involved in hydrogen bond interactions with most of the potential lead compounds throughout the MD simulations, which is consistent with the interaction profiles from the molecular docking study.

### 2.7. Evaluating Potential Leads via MM/PBSA Calculation

The MM/PBSA method has become a more efficient and reliable approach to model protein–ligand interactions. MM/PBSA provides reasonable approximations for free energy calculations and is more reliable than the conventional molecular docking process, yielding higher enrichment factors than docking [142]. MM/PBSA helps to prioritize compounds for experimental testing [143]. The Gibbs free energy of binding (ΔG_(bind)_) can be determined using Equation (1) [144,145].
ΔG_(bind)_ = ΔG_(complex)_ − [ΔG_(receptor)_ + ΔG_(ligand)_],(1)
where ∆G_(complex)_, ∆G_(protein)_ and ∆G_(ligand)_ are the total free energies of the protein–ligand complex, protein, and ligand, respectively.

Herein, the MM/PBSA approach was employed to determine the binding free energies as well as the energy contributions per-residue of the ADAR2-ligand complexes. Ligand structures of the top nine potential lead compounds and IHP are presented in Figure 8, while those of the other compounds listed in Table 2 are shown in Appendix A. The other contributing energy terms, including van der Waals (vdW), electrostatic, polar solvation and solvent accessible surface area (SASA) energies were also computed (Table 4). The vdW energy of the complexes ranged between −138 and −177.93 with the ADAR2-IHP having the highest vdW energy (−138.816 kJ/mol) [Table 4]. ZINC000085734971 demonstrated the most negative vdW energy with −177.923 kJ/mol, followed by ZINC000013462928 (−177.25 kJ/mol) and ZINC000085996580 (−174.655 kJ/mol) [Table 4]. For SASA energy, the values ranged from −14.949 to −22.88 kJ/mol with ZINC000085996580 and ZINC000100513617 demonstrating the most and least negative energy values, respectively (Table 4). SASA is linearly related to the non-polar solvation energy [144,146,147]. The SASA term is usually small and has little variations among similar ligands [144].

The ADAR2-IHP complex had a binding energy of −873.873 kJ/mol (Table 4). This is expected since IHP is a strong binder of ADAR2. Previous studies have shown that RNA binding neither alters the hydrogen bond network nor the binding of IHP in the active site of ADAR2 [27,38]. Interestingly, ZINC000085511995 demonstrated a more negative binding energy to the ADAR2 than IHP, with an energy value of −1068.26 kJ/mol (Table 4). ZINC000085511995 may function as an ADAR2 antagonist by first displacing IHP (an ADAR2 agonist) due to its more negative binding energy, which is suggestive of a stronger attraction. Naloxone, a competitive antagonist of opiate receptors, functions in a similar manner by selectively displacing agonists, such as morphine, thereby reversing their actions [148,149]. ZINC000044417732, ZINC000085850673, ZINC000014612330, and ZINC000100513617 also had comparable binding free energies of −642.856, −650.863, −648.56, and −567.619 kJ/mol, respectively (Table 4). The relatively good binding energy values of the potential lead compounds warrant the development and implementation of a low throughput functional assay complemented by biochemical assays for GluA2 and 5-HT2CR, among others, in a neuronal culture system to ascertain their potency for treating various neurological disorders.

#### Per-Residue Energy Decomposition

The energetic contribution of each residue to the interaction with the shortlisted compounds was investigated using the per-residue energy decomposition via g_mmpbsa [146]. Understanding the binding modes of proteins with other molecules and elucidating the hot-spot residues (residues directly involved in the interactions) could help in designing drugs to disrupt proteins [150,151]. Hotspot residues have been suggested to contribute more than 1 or 2 kcal/mol in protein-protein complexes [152]. However, for protein—ligand complexes, hot-spot residues have been suggested to contribute energies greater than 5 kJ/mol or less than −5 kJ/mol [153]. 

From the molecular interaction profiles, Arg400, Arg401, Lys519, Ser531, Leu532, Trp687, Glu689, and Lys690 were predicted to be critical in ligand binding. Therefore, we sought to investigate their importance in ligand binding via the energy decomposition method. For the ADAR2-IHP complex, several residues contributed more than the ±5 kJ/mol threshold. A total of 53 residues contributed favorably (less than −5 kJ/mol) to IHP binding. Arg400, Arg401, Lys519, Ser531, Leu532, Trp687, Glu689, and Lys690 contributed −51.4229, −73.4334, −74.3903, −1.9345, −2.1645, −3.9056, 113.0625, and −83.6689 kJ/mol, respectively. Other residues worth mentioning are Arg522, Lys629, Lys662, Lys672 and Asp695 which contributed −50.0431, −88.4847, −63.9655, −56.4605 and 91.1192 kJ/mol, respectively, to IHP binding. This result corroborates previous studies which report that IHP strongly and stably binds in the active site of ADAR2 and is not influenced by RNA binding to the ADAR2 [27,38]. The extremely basic nature of the ADAR2 active site favors IHP binding which in turn stabilizes multiple lysine and arginine residues [36,38].

The ADAR2-ZINC000085511995 complex, which had a binding free energy of −1068.26 kJ/mol, experienced a lot of residues contributing energies above the ±5 kJ/mol threshold, comparable to the ADAR2-IHP complex. A total of 54 residues contributed favorably to its binding in the active site of the ADAR2. The predicted critical residues comprising Arg400, Arg401, Lys519, Ser531, Leu532, Trp687, Glu689, and Lys690 had energies of −49.6330, −52.8445, −63.4549, −3.3215, −3.9087, −6.4576, 71.2105, and −83.5814 kJ/mol, respectively. The multiple residues and their contribution energies involved in ZINC000085511995 binding make ZINC000085511995 an interesting drug candidate to consider for ADAR2.

The ADAR2-ZINC000044417732 complex also had several residues contributing above the threshold. The critical residues, Arg400, Arg401, Lys519, Ser531, Leu532, Trp687, Glu689, and Lys690 were observed to contribute energies of −35.9893, −50.7197, −38.7647, −1.5284, −7.1948, −2.1630, 51.3166, and −34.4276 kJ/mol, respectively. For the ADAR2-ZINC000085950180 complex, only residues Arg400, Arg401, Leu404, Glu509, Lys519, Arg522, Leu532, Lys629, Leu632, Arg635, Lys662, Lys672, Glu689, Lys690, and Asp695 contributed above the threshold with energies of 6.9611, −8.1196, −6.0607, 8.0841, −5.3107, −7.2416, −6.6838, −26.3468, −9.2015, −6.0062, −10.7480, −8.1484, 28.8698, −9.8404, and 30.7624 kJ/mol, respectively. From the per-residue energy decomposition, residues Arg400, Arg401, Lys519, Trp687, Glu689 and Lys690 seem to be very crucial in ligand binding in the IHP binding site of ADAR2 than Ser531 and Leu532. Future ADAR2 drug initiatives may consider developing drugs with higher specificity to these residues.

### 2.8. Re-Docking Predicted Hits against 5-HT2C Receptor

Ritanserin had the most negative binding energy of −12.7 kcal/mol followed by ZINC000085996580 and ZINC000085850673 with binding energies of −11.5 and −10.9 kcal/mol, respectively (Appendix A). Ritanserin is a serotonin receptor antagonist, an antidepressant and has anxiolytic properties. Ritanserin is also an antiparkinsonian agent and has been shown to improve sleep [154,155,156]. Ritanserin improves motor deficits, including akinesia in mouse and humans [157]. Significant success was recorded when ritanserin (average dose of 13.5 mg/day) was administered to 10 patients with neuroleptic-induced Parkinsonism for 2 to 4 days. Out of the 10 patients with akathisia, 8 responded positively, observing a drop in Hillside Akathisia Scale (HAS) baseline ratings from 16.4 (± 6) to 7.4 (± 5.2) after 3 days of treatment [158]. However, ritanserin has not been approved for medical use due to safety and toxicity concerns, although it is widely used for research purposes [159,160,161]. Thus, there is the need to develop more potent and less toxic biomolecules that target serotonin receptors.

ZINC000085734971 had a binding energy of −10.6 kcal/mol, while both ZINC000085511995 and ZINC000085950180 had −10.5 kcal/mol. Both ZINC000044417732 and ZINC000014612330 also had binding energies of −10.3 kcal/mol while ZINC000013462928 and ZINC000100513617 had binding energies of −9.9 and −9.6 kcal/mol, respectively (Appendix A). A previous study docked compounds with the ethyl 2-(*p*-tolyloxy)acetate skeleton against the 5-HT2CR (PDB ID: 6BQH) using AutoDock Vina and the compound with the best affinity had a binding energy of −6.2 kcal/mol [162]. The strong binding affinities of the predicted leads could make them potential 5-HT2CR modulators. The PASS predictions and safety profiles of the identified molecules warrant further experimental studies to corroborate or validate their anti-ADAR2 and anti-schizophrenia properties. Further functional assays, both in a reduced system and in cultured neurons, are required to determine with greater resolution the relative efficacy of the predicted compounds on ADAR2, 5-HT2CR and GluA2.

### 2.9. Origin and Source of the Potential Lead Compounds 

The natural sources of the nine potential lead compounds were investigated from various databases, including ZINC15 [163], PubChem [164,165], ChEMBL [166,167,168], LOTUS [169], KNApSAcK [170,171], Natural Product Activity and Species Source (NPASS) [172], and Indian Medicinal Plants, Phytochemistry And Therapeutics (IMPPAT) [173] as previously conducted [174]. The pharmacological activities of the plant sources were also probed from the existing literature.

ZINC000044417732 (chitranone) has been isolated from the roots of *Plumbago zeylanica* [175,176] and *P. capensis* [177], and stem bark and fruits of *Diospyros maritima* [178,179,180] and *D. kaki* [180]. Chitranone has been previously reported to be cytotoxic against four cancer cell lines including human oral epidermoid carcinoma (KB), human lung cancer (Lu1), hormone-dependent human prostate cancer (LNCaP), and human umbilical vein endothelial cells (HUVEC) at EC_50_ values of 0.3, 1.1, 2.2, and 2.1 μg/mL, respectively [178]. The antimicrobial activity of chitranone has also been reported previously [178]. ZINC000085950180 (isozeylanone) is also found in *D. maritima* [179,180], *P. europaea* [181], and *P. zeylanica* [182]. Organic extracts of *P. zeylanica* were shown to have strong bactericidal activity against *Helicobacter pylori,* a type of bacteria associated with gastric cancer and peptic ulceration [183].

ZINC000085511995 (mamegakinone) has also been isolated from various *Diospyros* spp. [184], includng *D. batocana* [185], *D. obliquifolia* [186], *D. mollis* [184], *D. chamaethamnus* [184], *D. kaki* [187], *D. lotus* [188,189], *D. lycioides* [186], *D. montana* [189], *D. usambarensis* [190], and *D. zombensis* [184]. Additionally, ZINC000085850673 (3,8’-Bi[2-methyl-5-hydroxy-1,4-naphthoquinone] or 3,8’-biplumbagin) can be found in the fruit extract of *D. maritima* [179,180]. ZINC000085734971 (3,3’-Ethylidenebis(2-methyl-5-hydroxy-1,4-naphthoquinone) or ethylidene-3,3′-biplumbagin) can be found in extracts of *D. maritima* [179,180]. *D. kaki* has been used to treat patients with mental and physical complaints caused by trauma due to war, disaster, and burning [191].

ZINC000085996580 (lespedezol B2 or 8-[[2-(2,4-dihydroxyphenyl)-6-hydroxy-1-benzofuran-3-yl]methyl]-6H-[1]benzofuro[3,2-c]chromene-3,9-diol) has been extracted from the stem of *Lespedeza homoloba* [192]. Compounds derived from *L. homoloba* were found to exhibit strong anti-oxidative activity in the rat brain homogenate test against lipid peroxidation [192]. ZINC000014612330 ((4aR,12bR)-9-hydroxy-2,5,5-trimethyl-3,4,4a,12b-tetrahydronaphtho[3,2-c]isochromene-7,12-dione) and pyranokunthone A are isomers. Pyranokunthone A, obtained from the root bark of *Stereospermum kunthianum*, demonstrated moderate activity against two strains of *Plasmodium falciparum* comprising chloroquine-sensitive strain (IC_50_ of 11.7 μg/mL) and chloroquine-resistant clone (IC_50_ > 25 μg/mL) while having insignificant toxicity on the endothelial cell line ECV-304 (IC_50_ > 200 μg/mL) [193].

ZINC000100513617 (indigo) is found in the leaves and seeds of both *Isatis tinctoria* [194] and *I. indigotica* [195], fruits of *Couroupita guianensis* [196], and leaves of *Eupatorium laeve* [197]. Indigo and indirubin are widely known for their anti-inflammatory properties as they have been used in Central Europe since ancient times [194,198]. Bisindigotin, a derivative of indigo and indirubin, demonstrated a dose-dependent (0.1–5 μM) inhibition of 2,3,7,8-tetrachlorodibenzo-p-dioxin (TCDD)-induced ethoxyresorufin O-deethylase (EROD) activity in human HepG2 hepatoma cells with an IC_50_ of 0.8 μM [195]. In addition to its antipyretic, antiviral, anti-inflammatory, and anti-endotoxin properties, *I. indigotica* leaf extracts inhibit human hepatoma cell growth [199]. Moreover, several compounds isolated from *I. indgotica* and *tinctoria* have shown neuroprotective activities [200,201,202,203,204], thereby warranting the experimental testing of indigo.

ZINC000013462928 (isohibalactone or BDBM512896 or 3′,4′-bis(methylenedioxy)-lign-7(E)-en-9,9′-olide) can be obtained from *Hypoestes purpurea* [205], *Linum corymbulosum* [206], and leaves of *Juniperus chinensis* [207]. 3’,4’-bis(methylenedioxy)-lign-7(E)-en-9,9’-olide has a Z isomer [206] which is similar to cubebin. The anti-cancer, trypanocidal, anti-inflammatory, analgesic, anti-proliferative and leishmanicidal activities of cubebin have been highlighted in the literature [208,209]. Moreover, cubebin has been reported to have neuroprotective properties, as pretreatment of mice with cubebin (25 and 50 mg/kg) prevented scopolamine-induced learning and memory impairments [210]. Also, cubebin inhibited acetylcholinesterase with an IC_50_ of 992 µM [210], making ZINC000013462928 an interesting candidate to investigate for its neuroprotective and anti-ADAR2 property.

Some of the topmost compounds, including ZINC000085511995, ZINC000085950180, ZINC000044417732, ZINC000085850673, ZINC000085996580, and ZINC000085734971 are polyphenols. Polyphenols are known for their antioxidant [211,212], anti-inflammatory [211,213], and anti-cancer properties [214,215,216]. They also possess neuroprotective effects and help improve cognitive function [217,218,219]. However, similar to all bioactive compounds, polyphenols also have off-target interactions. One major off-target interaction of polyphenols is their ability to bind and inhibit the activity of certain drug metabolizing enzymes [220,221]. Some polyphenols have also been reported to modulate estrogen receptors and may exhibit estrogen-like effects [222,223,224,225]. Polyphenols also have the ability to interfere with nutrient absorption and metabolism in the body, especially, chelating iron and zinc, thereby inhibiting their absorption [226,227]. While this can be beneficial in preventing iron overload in patients with hemochromatosis, it can also lead to iron deficiency anemia in individuals with poor iron status.

Furthermore, polyphenols can also interact with the gut microbiota [228,229]. Some polyphenols have been found to promote the growth of beneficial gut bacteria, thereby improving gut health and immune function [230,231]. However, some polyphenols can also inhibit the growth of certain pathogenic bacteria, which can alter the gut microbiota composition and lead to negative effects [232,233]. Notwithstanding this, metal-phenolic networks (MPNs) can help with precise drug delivery, improve the efficacy, and limit the off-target toxicity of polyphenols as MPNs have negligible cytotoxicity [234].

## 3. Materials and Methods

Structure-based virtual screening (SBVS) was employed to identify potentially good ADAR2 binders with ADAR inhibitory activity (Figure 9). The study used IHP as a control, which is a known binder and agonist of the ADAR2 [27,38]. Two widely used molecular docking programs, AutoDock Vina and Maestro, were used for the molecular docking runs [47,235,236,237]. The top compounds found in both screening outputs were selected for further analysis. The molecular interactions between the ADAR2 and the novel potential inhibitors were determined (Figure 9). Furthermore, pharmacokinetics and toxicity profiling of the shortlisted compounds were performed to assess their drug-likeness. The biological activities of the compounds were then predicted using the Bayesian approach. The top potential lead compounds and the known binders were then subjected to molecular dynamics simulations including the MM/PBSA computations (Figure 9).

### 3.1. Obtaining and Preparing Protein and Ligand Structures

The hADAR2 structure was retrieved from the Research Collaboratory for Structural Bioinformatics Protein Data Bank (RCSB PDB) with corresponding PDB ID of 5ed2 [238,239]. The 5ed2 comprises a human ADAR2 protein structure complex with inositol hexakisphosphate (also known as phytic acid, myo-inositol hexakisphosphate, InsP6, IP_6_ or IHP), bound to a double-stranded ribonucleic acid (dsRNA) [27]. The dsRNA, ligand (IHP) and the Zinc atoms which were bound to the hADAR2 protein were removed using PyMOL (version 2.3.0). The resulting structure was processed using the Protein Preparation Wizard in Maestro (Schrödinger, LLC, New York, NY, USA). Additionally, the OPLS4 force field was employed to optimize protein energies and to remove any steric hindrance [240].

A total of 35,161 natural products from the Traditional Chinese Medicine (TCM) database were obtained from TCM@Taiwan, the world’s largest non-commercial TCM database, which is a catalog of ZINC15 database [163,241]. The compounds were pre-filtered based on molecular weight as previously conducted [242]. Compounds with molecular weights below 150 g/mol and above 600 g/mol were eliminated. A total of 25,196 compounds had molecular weights within the threshold and were used for the study. The ADAR2-bound ligand (IHP) was extracted from the 5ed2 structure and used in the study as a standard or control.

Ligand structures were prepared using the Ligand Preparation Wizard (LigPrep) in Maestro. For the TCM library, a total of 37,398 compounds (some being conformers of the 25,196 pre-filtered library) were generated using LigPrep. Various ionic states, tautomeric states and stereo chemistries were generated from each input ligand molecule at a pH of 7.0 ± 2.0 using Epik [243]. LigPrep optionally expands tautomeric and ionization states, ring conformations, and stereoisomers to generate chemical diversity from the input structures, accounting for the increase in number of ligands. LigPrep also optimizes the output structures for molecular docking in Maestro. For the bound ligand, IHP, 3 conformers were generated.

### 3.2. Determining Binding Sites

A literature search was conducted to identify previously reported active sites of the human ADAR2. Moreover, CASTp 3.0 (http://sts.bioe.uic.edu/castp/calculation.html accessed on 14 July 2022) [244] was employed to predict potential binding cavities of the hADAR2. Binding site predictions which had no openings or were relatively small for ligands to dock into were not considered.

### 3.3. Molecular Docking

Two molecular docking applications comprising Xglide (Glide Cross Docking) module in Maestro [47,235] and AutoDock Vina [237] were employed to screen natural products-derived compounds against the ADAR2 protein. Molecular docking tools tend to differ in their search and scoring algorithms, therefore, combining multiple docking tools compensates for individual limitations [245,246,247,248]. A total of 25,196 pre-filtered TCM compounds and IHP as control were used as the screening library.

For each docking tool, the top 10% (2520 compounds) of the total number of compounds were shortlisted after the docking process based on the docking scores. Traditional consensus scoring method was used to shortlist suitable compounds. In molecular docking, choosing compounds based on consensus scoring from a variety of docking programs produces a far higher predictive performance than using the docking scores from a single docking program [44]. Thus, compounds which were found in the top 10% for both docking tools were selected for further analysis.

#### 3.3.1. Validation of Molecular Docking Protocols

Inhibitor identification has been greatly aided by molecular docking simulations, which have allowed the prediction and investigation of potential protein–ligand interactions [249]. However, the pose quality prediction of the available docking tools must be assessed in order to attain higher performance. Thus, this study validated the ability of the two docking programs to accurately predict conformations similar to the experimental scenario as previously conducted [45,250,251,252]. The IHP was docked against the ADAR2 protein using both docking tools and the pose with the most negative binding energy from each tool was structurally aligned with the conformation in the crystallographic structure using the rigid module of LS-Align [253]. The RMSD values of the alignments were then determined.

#### 3.3.2. Molecular Docking via AutoDock Vina

The first step of the molecular docking run was performed using the AutoDock Vina module in PyRx (version 0.9.2) [236,237]. The energy of the 25,196 pre-filtered TCM compounds were minimized using the UFF force field, out of which 25,189 were successfully converted to AutoDock’s compatible format, Protein Data Bank, Partial Charge (Q), and Atom Type (T) (PDBQT). The docking process was performed with exhaustiveness set to 8, using grid box dimensions of 64.306 × 53.098 × 51.413 Å^3^ and the protein centered at 30.697 Å, 27.166 Å and 82.548 Å.

#### 3.3.3. Molecular Docking via XGlide (Maestro)

For the second phase of the molecular docking process, Glide Cross Docking (XGlide) was employed [47,235]. For the grid generation, SiteMap [254] was used to locate active sites which were used to determine the grid box. Default inner (10 × 10 × 10 Å^3^) and outer box sizes (26 × 26 × 26 Å^3^) were used for the virtual screening. Ligand vdW scale factor and cut-off for a ‘good’ RMSD were set to default (0.80 and 2.0 Å, respectively). The Standard Precision (SP) algorithm was used for the molecular docking run [235]. For compounds with more than one tautomer, the docking pose with the most negative binding energy was selected after the molecular docking process.

#### 3.3.4. Shortlisting Compounds using Consensus Scoring

To facilitate compound selection, the consensus scoring approach was used. A previous study had success rates of ~82% when multiple docking tools, including AutoDock [255], AutoDock Vina [237] and DOCK6 [256], were combined to identify inhibitors [44]. The accuracies of the individual docking tools were 55%, 64% and 58%, respectively [44], implying a far greater success rate when multiple docking tools are used. The top 10% (2520) of the initial pre-filtered library were shortlisted based on binding energy for further studies. Traditional consensus scoring was employed to rank the top compounds [257]—the consensus score for each compound was obtained by averaging the docking scores for both AutoDock Vina and Maestro (Glide).

### 3.4. Determining the Interactions between the ADAR2-Ligand Complexes

Protein–ligand interactions help us to understand the mechanisms of ligand binding, which is crucial in drug design and development [258]. Ligand binding via specific molecular recognition can trigger protein activation or inactivation and affect protein function [259]. Thus, elucidating the mechanisms of binding and the residues involved in these interactions is relevant for drug design. This study therefore visualized the molecular interactions between the hADAR2 protein and docked ligands (selected compounds) using Maestro.

### 3.5. Determining ADMET Properties

The pharmacokinetic profiles comprising absorption, distribution, metabolism, excretion (ADME) of the shortlisted compounds were determined using SwissADME [74]. OSIRIS Datawarrior 5.5.0 was also used to determine the toxicity risks of the compounds by predicting their mutagenicity, tumorigenicity, irritancy and reproductive effects [85,260]. Compounds which were predicted to have more than one toxicity risk were ignored in this study. Additionally, compounds that were predicted to be either tumorigenic or mutagenic were eliminated.

### 3.6. Structural Similarity Search and Prediction of Biological Activity of Compounds

Prediction of Activity Spectra of Substances (PASS) was employed to predict the biological activity of the identified compounds [91,92,93]. PASS, with average prediction accuracy of ~95% based on leave-one-out cross validation (LOO CV) estimation, uses a Bayesian approach to predict the possible biological activity of a molecule based on its structure [92]. Furthermore, similarity search of the shortlisted compounds was performed via DrugBank [261,262] in order to identify structural analogs with relevant biological activity.

### 3.7. Molecular Dynamics Simulations

The unbound protein and selected ADAR2-ligand complexes were subjected to 100 ns molecular dynamics (MD) simulations using the GROningen MAchine for Chemical Simulations (GROMACS) version 5.1.5. Ligand topologies of the compounds for OPLS force field were generated using the LigParGen [263]. Each system was solvated in a cubic box using the “TIP4P” water model and the OPLS/AA force field [264,265]. Either sodium or chlorine ions were added to each system to neutralize the charges. The systems were then subjected to constant-number, constant-volume and constant-temperature (NVT) and isothermal-isobaric or constant-number, constant-pressure and constant-temperature (NPT) ensembles, prior to the MD simulation. The RMSD, Rg and RMSF of each system were analyzed after the MD simulations. Moreover, the number of hydrogen bonds during the simulation was assessed for each system. Snapshots at 25 ns intervals (time step = 0, 25, 50, 75 and 100 ns) were generated for each complex.

### 3.8. Molecular Mechanics Poisson–Boltzmann Surface Area (MM/PBSA) Computation of Potential Leads

The binding free energies and the other energy components including vdW, electrostatic, polar solvation and solvent accessible surface area (SASA) energies of each complex were determined using the MM/PBSA method via the g_mmpbsa tool [146]. The energetic contribution per-residue to ligand binding in the IHP binding site was also evaluated [146].

### 3.9. Re-Docking Potential Leads against the 5-HT2CR

The identified potential leads were finally screened against the 5-Hydroxytryptamine (or serotonin) 2C receptor (5-HT2CR) to evaluate their binding to the receptor using AutoDock Vina. The 3D structure of 5-HT2CR in complex with ritanserin was retrieved from RCSB PDB with corresponding ID 6BQH [266]. Ritanserin, which is a known antagonist and a strong binder of 5-HT2CR, was extracted from the complex and used as the control for the molecular docking. Molecular docking was performed using the ritanserin binding site with docking grid box dimensions of 25.0 × 25.0 × 43.0801246543 Å^3^ and the protein centered at 40.5731 Å, 33.1458 Å and 52.5793299511 Å.

## 4. Conclusions

This study carefully predicted potential ADAR2 inhibitors which can be beneficial in the treatment of various diseases including some cancers, viral infections and neurological disorders. We employed two widely used molecular docking tools, namely AutoDock Vina and Glide (Maestro, Schrodinger Suite) to predict the binding affinities of natural compounds from the Chinese flora and fauna. The top nine shortlisted compounds had favorable binding to the human ADAR2 and 5-HT2C receptor and were predicted to have insignificant toxicities. They also had favorable results when subjected to MD simulations and MM/PBSA calculations. One of the potential leads, ZINC000085511995, had higher binding free energy (−1068.26 kJ/mol) to the ADAR2 than the known binder, IHP (−873.873 kJ/mol), after MM/PBSA computations. The biological activity predictions and structural similarity search also corroborated the activity of the predicted compounds. The identified biomolecules can help accelerate the pace of ADAR2 research since they can be potential tools for understanding the biology of RNA editing and as a starting point for the development of potential therapeutic agents. Further experimental studies on the predicted molecules are required to determine their anti-ADAR2 activity. This study also observed the presence of naphthoquinone, indole, furanocoumarin and benzofuran moieties in most of the top compounds. Serotonin and tryptophan, which are beneficial in digestive regulation, sleep cycle and improved mood, also have the indole moiety [48,49,53,55]. Moreover, vitamin K, a naphthoquinone derivative, has been reported to improve cognitive function [122,123]. The identified chemical series are worthy of further experimental testing to ascertain their potency for extensive medicinal chemistry and biological characterization. These series may give rise to new putative ADAR2 inhibitors which can be designed using de novo/rational design, virtual focused combinatorial libraries (vFCL) generation, and in silico screening of the vFCL.

## Figures and Tables

**Figure 1 ijms-24-06795-f001:**
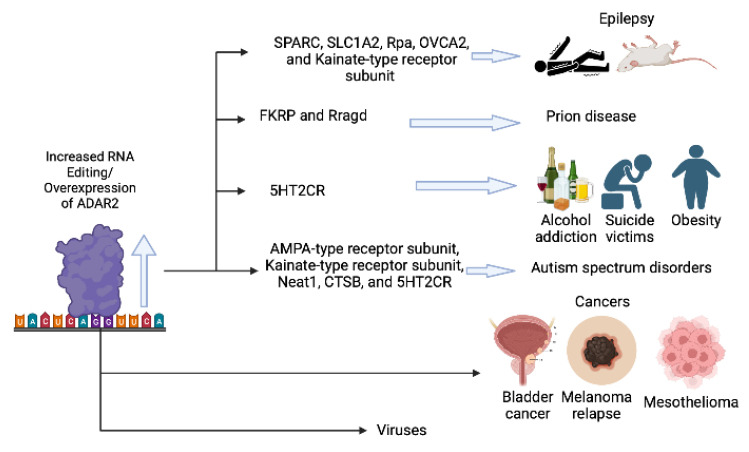
Schema representing the mechanism of ADAR2 overexpression (or increased RNA editing) and the associated diseases. The figure was created using BioRender (https://biorender.com/ accessed on 30 March 2023).

**Figure 2 ijms-24-06795-f002:**
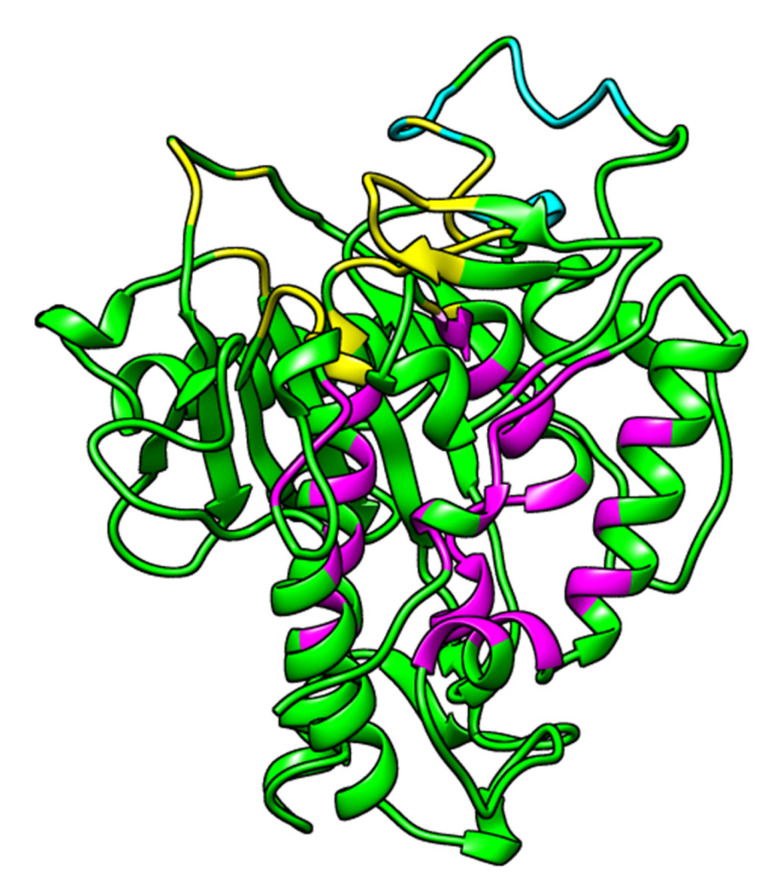
3D structure of ADAR2 (PDB ID 5ED2) showing the predicted binding sites. Predicted binding sites are colored yellow, magenta and cyan for pockets 1, 2 and 3, respectively. Image was generated using UCSF Chimera version 1.16 [42].

**Figure 3 ijms-24-06795-f003:**
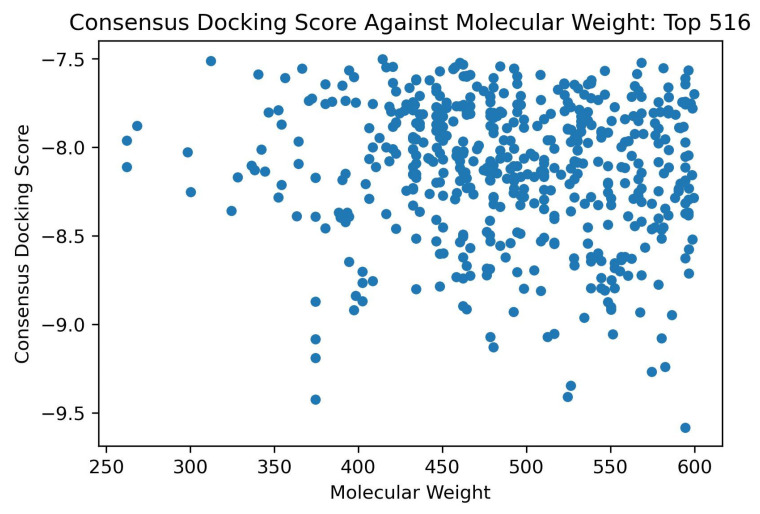
Consensus docking score against the molecular weight plot of the top 516 shortlisted compounds from molecular docking.

**Figure 4 ijms-24-06795-f004:**
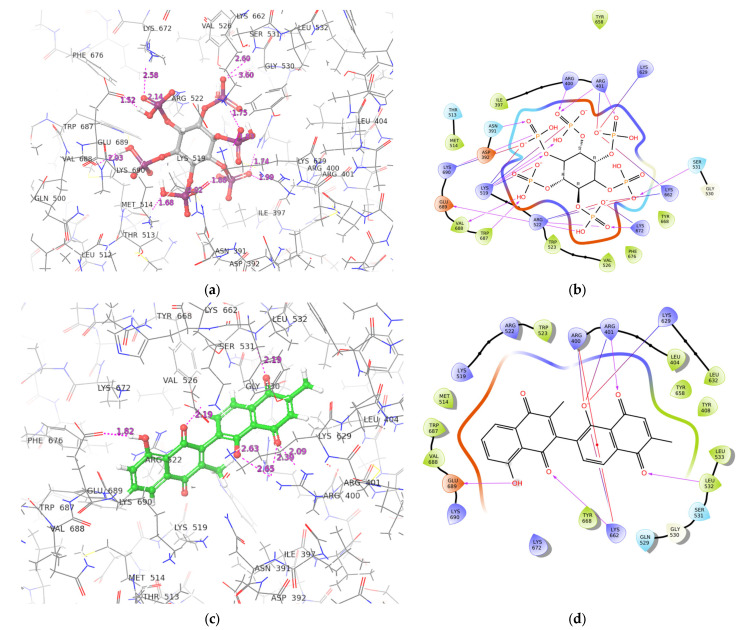
Conformation and protein–ligand interaction profiles of the binding pose with the most negative binding energy of IHP (**a**,**b**) and ZINC000044417732 (**c**,**d**). For the interaction profiles, purple arrows, red lines and combination of “red and blue” lines represent hydrogen bonds, pi-cation interactions and salt-brides, respectively.

**Figure 5 ijms-24-06795-f005:**
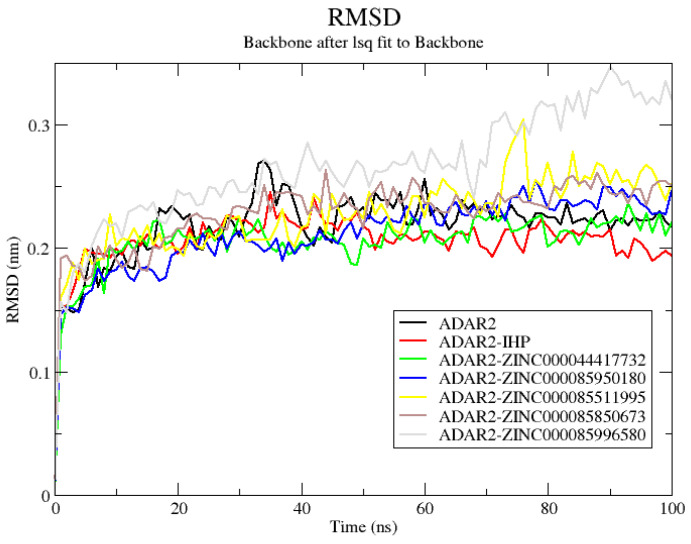
RMSD plot of the unbound ADAR2 and ADAR2-ligand complexes. The unbound protein, ADAR2-IHP, ZINC000044417732, ZINC000085950180, ZINC000085511995, ZINC000085850673, and ZINC000085996580 complexes are colored black, red, green, blue, yellow, brown, and grey, respectively.

**Figure 6 ijms-24-06795-f006:**
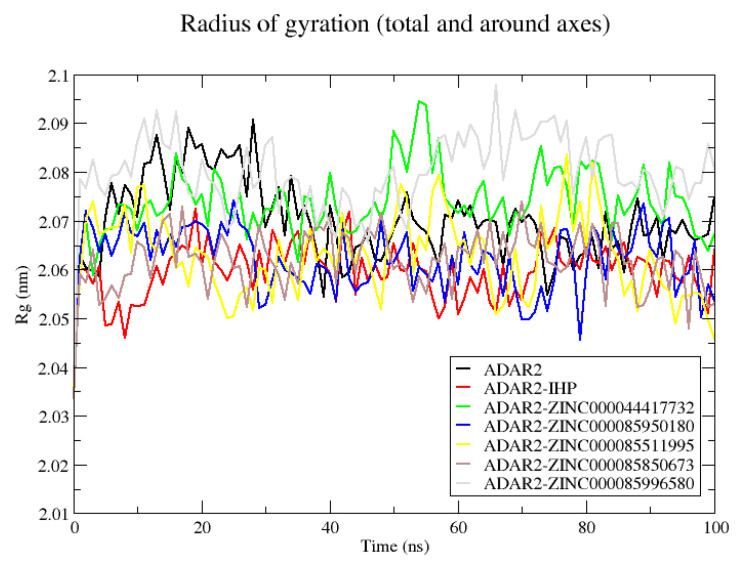
Radius of gyration plot of the unbound ADAR2 and ADAR2-ligand complexes. The unbound protein, ADAR2-IHP, ZINC000044417732, ZINC000085950180, ZINC000085511995, ZINC000085850673, and ZINC000085996580 complexes are colored black, red, green, blue, yellow, brown, and grey, respectively.

**Figure 7 ijms-24-06795-f007:**
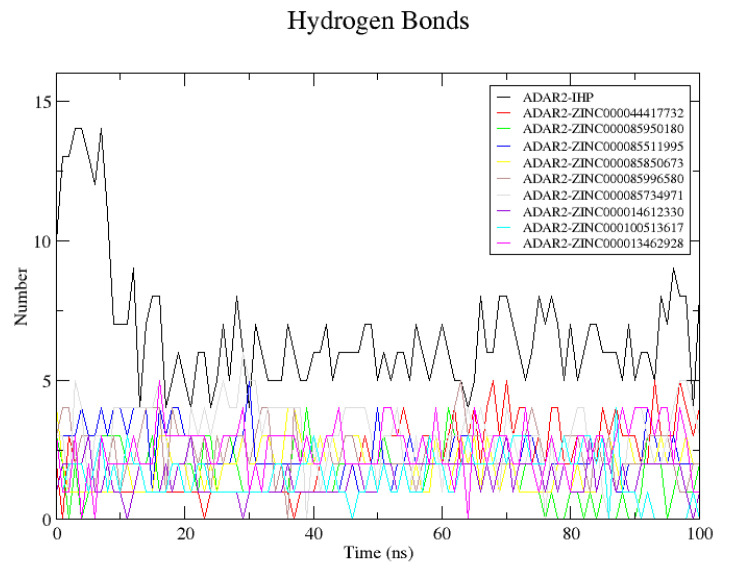
Hydrogen bond analyses of each ADAR2-ligand complex throughout the 100 ns MD simulation.

**Figure 8 ijms-24-06795-f008:**
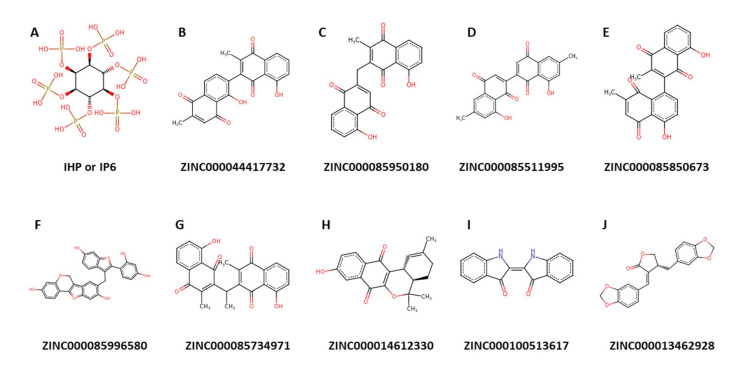
Chemical structures of IHP and the nine potential lead compounds.

**Figure 9 ijms-24-06795-f009:**
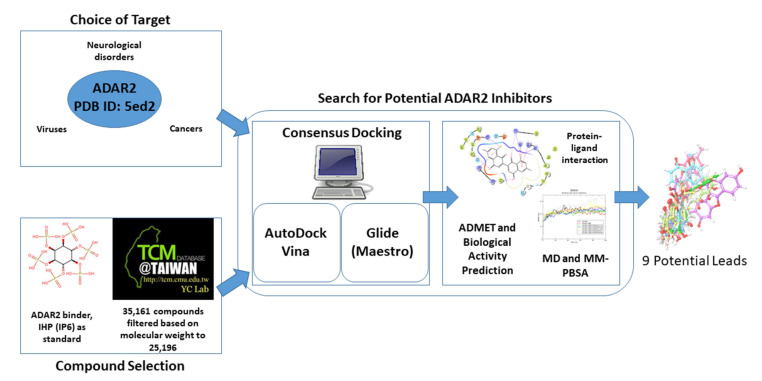
Schema detailing the step-by-step approach employed in this study to identify novel potential ADAR2 inhibitors.

**Table 1 ijms-24-06795-t001:** Predicted binding cavities via CASTp with their area, volumes and residues lining each pockets.

Pocket No.	Area (Å^2^)	Volume (Å^3^)	Residues Lining the Pocket
1	397.972	455.471	Lys350, Val351, Gly374, Thr375, Lys376, Cys377, Ile378, Asn379, His394, Ala395, Glu396, Ile446, Thr448, Ser449, Pro450, Cys451, Gly452, Arg455, Ile456, Pro459, Lys483, Ile484, Glu485, Ser486, Gly487, Gln488, Gly489, Thr490, Leu511, Thr513, Cys516, Arg590, Lys594, and Ala595.
2	541.281	342.976	Ala389, Leu390, Asn391, Asp392, Ile397, Arg400, Arg401, Leu404, Tyr408, Gln500, Leu512, Thr513, Met514, Lys519, Arg522, Trp523, Val526, Gly527, Ile528, Gln529, Gly530, Ser531, Leu532, Leu533, Lys629, Leu632, Tyr658, His659, Lys662, Leu663, Tyr668, Gln669, Lys672, Phe676, Trp687, Val688, Glu689, Lys690, Pro691, Thr692, Gln694, and Asp695.
3	120.812	129.477	Ser458, His460, Glu461, Pro462, Ile463, Glu466, Pro467, Ala468, Asp469, Arg470, His471, His552, Asp554, and His555.

**Table 2 ijms-24-06795-t002:** Binding energies of IHP and some selected compounds with ADAR2. The interacting residues are also shown. The glide docking and consensus scores presented here are rounded to 2 decimal places.

Compound	Binding Energy (kcal/mol)			Interacting Residues		
	AutoDock Vina	Glide	Consensus Score	H-Bond	Pi-Cation	Salt Bridges
ZINC000044417732	−10.9	−7.95	−9.42	Arg401, Leu532, Lys662 and Glu689	Arg400 and Lys662	Arg400, Arg401, Lys629 and Lys662
ZINC000085950180	−10.5	−7.88	−9.19	Arg401, Ser531 (2), Lys629, Trp687 and Asp695	Lys662	-
ZINC000085511995	−10.9	−7.27	−9.08	Arg522, Lys629 and Trp687	Arg400 and Lys662	Arg401, Lys519, Lys629, Lys662 and Lys690
ZINC000085850673	−10.5	−7.24	−8.87	Arg401, Arg522, Lys629, Tyr658 and Lys690	Lys519 and Lys629.	Lys519 and Lys690
ZINC000085996580	−11	−6.62	−8.81	Arg401 (2), Lys519 and Lys690	Arg400, Arg522 and Lys662 (4)	-
ZINC000085734971	−10.6	−6.93	−8.76	Arg401, Ser531, Lys629, Lys662 (2) and Asp695	Lys629	-
ZINC000034517814	−9.4	−8.11	−8.75	Met514, Arg522, Ser531, Glu689 (2) and Lys690	-	-
ZINC000014613520	−9.9	−7.39	−8.64	Arg401, Ser531, Lys662 and Trp687	Arg522 and Lys662	Arg522
ZINC000095911588	−9.4	−7.86	−8.63	Arg401, Ser531, Arg522, Trp523 and Lys629	Lys662	Arg400
ZINC000085569519	−10.1	−6.74	−8.42	Tyr408, Lys629, Tyr658, Lys662 and Lys690	-	-
ZINC000008234342	−9.7	−7.08	−8.39	Leu532	Arg400 and Lys662	Arg400, Arg401 and Lys629
ZINC000085569292	−9.3	−7.48	−8.39	Arg400 (2), Arg401 and Lys662	-	Lys519 and Lys690
ZINC000095911414	−10.1	−6.65	−8.38	Arg401 (2), Ser531, Lys629 and Trp687	Arg400 and Lys662	-
ZINC000086050572	−9.8	−6.94	−8.37	Arg401, Ser531, Lys629, Lys662 (2) and Asp695	-	-
ZINC000014612330	−10.2	−6.51	−8.36	Arg401, Ser531 and Lys629	-	-
ZINC000014814624	−10.0	−6.57	−8.28	Arg522, Ser531 and Lys690	Arg400, Arg522 and Lys662	Arg400, Arg401 and Lys629
ZINC000004098700	−9.0	−7.50	−8.25	Arg401 (2), Arg522 and Ser531	Arg400 and Lys662	Lys519
ZINC000095912516	−9.8	−6.57	−8.18	Arg401, Arg522, Leu532, Lys629, Tyr658 and Glu689	-	-
ZINC000085488788	−9.5	−6.84	−8.17	Arg401, Ser531 and Lys629	Lys519	Arg401, Lys629 (2) and Lys662
ZINC000070454227	−9.0	−7.34	−8.17	Arg401, Arg522, Ser531 and Lys690	Lys662	Lys519 and Lys690
IHP	−8.6	−7.97	−8.26	Asn391, Arg400, Arg401 (2), Lys519, Ser531, Lys672, Trp687, Val688, Glu689 and Lys690	-	Arg401, Lys519, Arg522, Lys629, Lys662, Lys672 and Lys690

**Table 3 ijms-24-06795-t003:** Pharmacokinetic evaluation of IHP, fluoxetine, nebularine, doxorubicin, and the 9 potential lead compounds. The consensus logP value (SwissADME) is reported in this table.

Compound	MW (g/mol)	logP o/w	TPSA (Å^2^)	BBB Permeant	GI Absorption	ESOL Solubility Class	No of Lipinski’s Rule Violations	No. of Veber’s Rule Violations
IHP	660.04	–6.77	459.42	No	Low	High	3	2
ZINC000044417732	374.34	2.88	108.74	No	High	Moderate	0	0
ZINC000085950180	374.34	2.71	108.74	No	High	Moderate	0	0
ZINC000085511995	374.34	2.82	108.74	No	High	Moderate	0	0
ZINC000085850673	374.34	2.9	108.74	No	High	Moderate	0	0
ZINC000085996580	508.48	4.47	136.66	No	Low	Poor	1	0
ZINC000085734971	402.4	3.39	108.74	Yes	High	Moderate	0	0
ZINC000014612330	324.37	3.05	63.6	Yes	High	Moderate	0	0
ZINC000100513617	262.26	2.16	58.2	Yes	High	Moderate	0	0
ZINC000013462928	352.34	3.22	63.22	No	High	Moderate	0	0
Fluoxetine	309.33	4.32	21.26	Yes	High	Moderate	0	0
Nebularine	252.23	−1.16	113.52	No	High	Very	0	0
Doxorubicin	543.52	0.44	206.07	No	Low	Soluble	3	1

**Table 4 ijms-24-06795-t004:** Contributing energy terms for the protein–ligand complexes determined via MM/PBSA calculations. The energy values are presented as “energy ± standard deviation”. All energy values are in kJ/mol.

Compound	vdW	Electrostatic Energy	Polar Solvation Energy	SASA Energy	Binding Energy
IHP	−138.816 ± 2.243	−1597.111 ± 7.065	883.140 ± 5.474	−20.866 ± 0.095	−873.873 ± 6.225
ZINC000044417732	−164.45 ± 1.303	−798.403 ± 4.671	340.231 ± 3.618	−19.937 ± 0.089	−642.856 ± 3.746
ZINC000085950180	−172.457 ± 1.278	−66.957 ± 2.459	123.492 ± 2.987	−19.151 ± 0.101	−135.075 ± 3.285
ZINC000085511995	−157.3 ± 1.578	−1550.773 ± 5.372	659.676 ± 5.323	−19.816 ± 0.108	−1068.26 ± 4.122
ZINC000085850673	−170.78 ± 1.79	−818.946 ± 5.622	357.645 ± 5.284	−18.914 ± 0.09	−650.863 ± 4.925
ZINC000085996580	−174.655 ± 1.248	−87.798 ± 1.888	203.977 ± 2.763	−22.88 ± 0.144	−81.304 ± 2.269
ZINC000085734971	−177.923 ± 1.177	−37.521 ± 2.675	141.105 ± 3.714	−20.808 ± 0.093	−95.133 ± 4.263
ZINC000014612330	−167.284 ± 1.304	−774.68 ± 2.258	310.493 ± 1.672	−17.154 ± 0.074	−648.56 ± 2.801
ZINC000100513617	−114.531 ± 1.602	−764.552 ± 3.591	326.488 ± 4.91	−14.949 ± 0.093	−567.619 ± 4.003
ZINC000013462928	−177.25 ± 1.238	−19.523 ± 1.104	56.374 ± 1.273	−18.245 ± 0.108	−158.661 ± 1.669

## Data Availability

Not applicable.

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
