# Peer review of "Molecular Docking and Dynamics Simulation Studies Predict Potential Anti-ADAR2 Inhibitors: Implications for the Treatment of Cancer, Neurological, Immunological and Infectious Diseases"

_ijms, 2023, doi:10.3390/ijms24076795_

Round 1

Reviewer 1 Report

The authors describe the screening a library of 35,161 compounds 34 obtained from traditional Chinese medicine against human ADAR2 is a potential therapeutic target for managing various disorders including autism and intellectual disability, in addition to depression, and schizophrenia. The authors utilized extensive molecular simulation protocol to apply consensus scoring and obtain valid ranking of the screened compounds. Two different docking modules including Maestro Glide , and Autodock Vina and Gromacs for molecular dynamics were used in the study. The study design, although only uses computational tools, is clear and rigorous and the results were well presented and discussed. The experimental part is informative and clear. The discussion part was well written, and the conclusion drawn was reliable. The manuscript is over all organized and uses appropriate language. Although the work impact looks moderate, I would expect it to get good volume of citations because of the research protocol design and justification that would address one of the biggest problem with solely in silico studies which is the results validation.

This work merits publication after minor corrections.

1.       The authors is giving details about basic information such as docking, dynamics,..etc. I understand that the authors would like to gain the highest number of citations for their manuscript, however some information are more suitable for reviews rather than a research article.

2.       The author is using a big number of references that increases the complexity and size of the manuscript without added value.

3.       The Authors did not mention the final number of hits in the abstract nor the conclusion.

4.       The author did not mention any previously known pharmacological activity for the potential leads discovered and whether any of them have been tested in any biological targets. The author should look up such information from the ZINC database or PubChem or any other relevant resources.

5.       Most of the authors have biochemistry, biology, molecular pharmacology, molecular pharmacology background, however, they did not support the presented research with any in vitro or in vivo testing to support their hypothesis. A minimal biological testing would greatly add to the impact of the manuscript.

Reviewer 2 Report

Dear Authors,

I have gone through your paper, and I have noticed some issues with the manuscript as described below:

# In the section of Introduction, I found that the authors have properly shown full forms of abbreviations.

# Moreover, it would be more meaningful, if authors could provide the schematic diagram for elaboration of concern mechanism of actions.

# Authors have forgotten to share details on CASTP server, please provide the link along with the date of accessions.

# For validations of docking protocol, there are some other literatures, which can be cited as: Medicine in Drug Discovery 2, 100008; Current Computer-Aided Drug Design 17 (2), 294-306

# Many fullforms such as RMSD has been missed from the manuscript.

# For Authors needs to give grid dimensions used for Autodock Vina and Glide docking protocols.

# Please improve Figure 1.

# Figure 2, Schrodinger Glide-based images are blurry with no interaction distance between ligand and amino acid residues.

# Obtained 9 hits should be preliminary tested for their in-vitro data,

Unless the author provides some experimental data, I can not recommend this publication.

# Simulation results needs to be improved, at current state, complexes are representing strong fluctuations in RMSD.

Reviewer 3 Report

The manuscript entitled "Molecular Docking and Dynamics Simulation Studies Predict Potential Anti-ADAR2 Inhibitors: Implications for the Treatment of Cancer, Neurological, Immunological and Infectious

Diseases" presented by Whelton A. Miller III et al. shows an interesting theoretical study relating a library of 35161 compounds to ADAR2 inhibition. The authors performed a detailed study to select the 9 most active compounds against various ADAR2-related diseases. publication of this manuscript in IJMS and I only recommend including in table 3 the data on Mutagenic, Tumorigenic, Irritant and Reproductive effect of table S1 of the 9 most important compounds (Figure 6)

Reviewer 4 Report

The authors employed molecular docking software and Molecular dynamics simulations to identify ligands of the human ADAR2. Then, they predicted these ligands' biological activity and ADMET properties. This study reported ADAR2 ligands as potential candidates for treating cancer and neurological, immunological, and infectious diseases.

Here, I have several questions and suggestions to the authors:

1, Could the authors offer more background introduction about the ADAR2 ligands?

2, Could the authors show the predicted binding pockets in the ADAR2 structure?

3, The docking poses shown in figure 2 are not clear. Could the authors highlight the key interactions?

4, Could the authors show the 2D chemical structures of compounds listed in Table 2?

5, Could the authors discuss the possible off-targets of polyphenols shown in the manuscript?

6, Could the authors provide positive and negative controls for predicting biological activity and ADMET properties?

Round 2

Reviewer 2 Report

Dear Authors, 

MS has been sufficiently revised now.

Reviewer 4 Report

The authors have answered all my questions. I think this manuscript is ready for publication.